# Pervasive epigenetic effects of *Drosophila* euchromatic transposable elements impact their evolution

**Yuh Chwen G Lee[1,2]\*, Gary H Karpen[1,2]\***

[1]Division of Biological Systems and Engineering, Lawrence Berkeley National Laboratory, Berkeley, United States; [2]Department of Molecular and Cell Biology, University of California Berkeley, Berkeley, United States

**Abstract** Transposable elements (TEs) are widespread genomic parasites, and their evolution has remained a critical question in evolutionary genomics. Here, we study the relatively unexplored *epigenetic* impacts of TEs and provide the first genome-wide quantification of such effects in *D. melanogaster* and *D. simulans*. Surprisingly, the spread of repressive epigenetic marks (histone H3K9me2) to nearby DNA occurs at >50% of euchromatic TEs, and can extend up to 20 kb. This results in differential epigenetic states of genic alleles and, in turn, selection against TEs. Interestingly, the lower TE content in *D. simulans* compared to *D. melanogaster* correlates with stronger epigenetic effects of TEs and higher levels of host genetic factors known to promote epigenetic silencing. Our study demonstrates that the epigenetic effects of euchromatic TEs, and host genetic factors modulating such effects, play a critical role in the evolution of TEs both within and between species.

**\*For correspondence:** grylee@lbl.gov (YCGL); GHKarpen@lbl.gov (GHK)

**Competing interests:** The authors declare that no competing interests exist.

## Introduction

Transposable elements (TEs) are genetic elements that can copy and transpose themselves into new genomic locations. Even though there are incidental reports of potentially adaptive TEs (*Daborn et al., 2002*; *Schlenke and Begun, 2004*; *Aminetzach et al., 2005*; *González et al., 2008*; *Schmidt et al., 2010*; *Mateo et al., 2014*; *Hof et al., 2016*), they generally lower host fitness (*Mackay, 1989*; *Pasyukova et al., 2004*) and are widely recognized as 'genomic parasites'. Despite the deleterious fitness consequences of TEs, they comprise appreciable and highly variable proportions of euchromatic genomes in all eukaryotes surveyed (*Biémont, 2010*; *Elliott and Gregory, 2015*; *Chalopin et al., 2015*). However, fundamental questions remain about the mechanisms that restrict the selfish increases in TE copy number and contribute to the wide variation of TEs within and between species.

Theoretical analyses predict that, in an outbreeding and meiotically recombining host population, copy number of TEs can be contained (i.e. reach an equilibrium) if the increase in copy number through transposition is counterbalanced by the removal of TEs (*Charlesworth and Charlesworth, 1983*; *Langley et al., 1983*). One possible mechanism for the containment of TEs is to regulate the transposition rate to equal the removal rate. Small RNAs in *Drosophila*, mammals, and plants are enriched for TE sequences and regulate the transposition of TEs in host germlines (*Girard et al., 2006*; *Gunawardane et al., 2007*; *Brennecke et al., 2007*; *Aravin et al., 2007*; *Slotkin et al., 2009*). These small RNAs guide the Ago and/or Piwi subfamilies of Argonaute proteins (reviewed in [*Hutvagner and Simard, 2008*]) to TE transcripts with complementary sequences, resulting in post-transcriptional silencing (reviewed in [*Klattenhoff and Theurkauf, 2008*; *Girard and Hannon, 2008*; *Senti and Brennecke, 2010*]). In addition, TEs can be transcriptionally silenced through small-RNA

**eLife digest** The DNA inside an organism encodes all the instructions needed for the organism to develop and work properly. Organisms carefully organize and maintain their DNA (collectively known as the genome) so that the genetic information remains intact and the cell can understand the instructions. However, there are some pieces of DNA that are capable of moving around the genome. For example, pieces known as transposable elements can make new copies of themselves and jump into new locations in the genome. Most transposons do not appear to have any important roles, and in fact they are usually harmful to organisms. Despite this, transposons are present in the genomes of almost all species. The number of transposons in a genome varies greatly between individuals and species, but it is not clear why this is the case.

Organisms have evolved ways to limit the damage caused by transposons. For example, many cells package regions of DNA containing transposons into a tightly packed structure known as heterochromatin. However, this type of DNA packaging sometimes spreads to neighboring sections of DNA. This is a problem because cells are not usually able to read the information contained within heterochromatin. This means that transposons can prevent some instructions from being produced when they should be. Lee and Karpen used fruit flies to investigate to what extent transposons harm organisms by changing the way DNA is packaged, and whether this influences how transposons evolve.

The experiments show that that more than half of the transposons in fruit flies cause neighboring sections of DNA to be packaged into heterochromatin. This can negatively impact up to 20% of genes in the genome. As a result, transposons that have harmful effects on DNA packaging are more likely to be lost from the fly population during evolution than transposons that do not have harmful effects. Fruit fly species containing transposons that tend to package more neighboring sections of DNA into heterochromatin generally have fewer transposons than genomes containing less harmful transposons.

The findings of Lee and Karpen provide new insight as to why the numbers of transposons vary among organisms. The next challenge is to find out whether transposons that alter how DNA is packaged are also common in primates and other animals.

guided enrichment of repressive epigenetic marks, which include DNA modifications (such as methylation) and post-translational histone modifications (such as di- and tri-methylation of H3 lysine 9 (H3K9me2/3), [*Klenov et al., 2007*; *Aravin et al., 2008*; *Sienski et al., 2012*; *Le Thomas et al., 2013*]). Both post-transcriptional and transcriptional silencing mechanisms reduce the RNA and protein output from TEs, and accordingly lower TE transposition rate. However, despite the presence of small-RNA regulation, measured transposition rates of TEs are significantly higher than excision rates (reviewed in [*Charlesworth and Langley, 1989*]). Furthermore, euchromatic TE insertions in multiple outbreeding species have low population frequencies (*Charlesworth and Langley, 1989*; *Dolgin et al., 2008*; *Lockton et al., 2008*; *González et al., 2008*; *Lockton and Gaut, 2010*; *Cridland et al., 2013*; *Kofler et al., 2015*), and a reduction in transposition rate alone is unlikely to explain such observations.

Alternatively, selection against the deleterious effects of TEs has been theoretically proposed and empirically supported as a major force that removes TE insertions from host populations and shapes the population dynamics of TEs (reviewed in [*Charlesworth and Langley, 1989*; *Lee and Langley, 2010*; *Barrón et al., 2014*]). It is well-established that TEs can be deleterious through their *genetic* effects, such as inserting into and disrupting genes and other functional elements (*Cooley et al., 1988*; *Finnegan, 1992*), acting as ectopic regulatory elements (*Feschotte, 2008*), and mediating ectopic recombination that results in detrimental chromosomal rearrangements (*Langley et al., 1988*; *Montgomery et al., 1991*; *Petrov et al., 2003*; *Mieczkowski et al., 2006*). On the other hand, TE insertions can also influence the epigenetic states of adjacent functional sequences, interfering with gene regulation ('epigenetic effects'; reviewed in [*Slotkin and Martienssen, 2007*]). A genome-wide study in *A. thaliana* first established the associations between DNA methylation of TEs and lower transcript levels of adjacent genes (*Hollister and Gaut, 2009*). Later studies identified TEs

as a major cause for DNA methylation-enriched regions in the *A. thaliana* genome (***Ahmed et al., 2011***; ***Schmitz et al., 2013***; ***Dubin et al., 2015***; ***Quadrana et al., 2016***; ***Kawakatsu et al., 2016***; ***Stuart et al., 2016***), and several demonstrated that this association results from spreading of DNA methylation from epigenetically silenced TEs (***Ahmed et al., 2011***; ***Quadrana et al., 2016***; ***Stuart et al., 2016***). Associations between TEs and enrichment of repressive epigenetic marks were also documented in mouse cell lines (***Rebollo et al., 2012***) and maize (***Eichten et al., 2012***; ***West et al., 2014***). However, only (***Hollister and Gaut, 2009***) explored the influences of these TE-induced enrichment of repressive epigenetic marks on the evolutionary dynamics of TEs.

In *Drosophila*, TE-induced enrichment of repressive epigenetic marks at functional elements in euchromatin was first solidly supported by comparing the epigenetic states of reporter genes in constructs with and without adjacent TEs (***Sentmanat and Elgin, 2012***). The same study also found that epigenetic effects of TEs depend on small-RNA targeting, and thus on host-directed transcriptional silencing of TEs. This spreading of repressive epigenetic marks from epigenetically silenced euchromatic TEs is reminiscent of the well-studied position-effect variegation (PEV), in which repressive epigenetic marks from pericentromeric or subtelomeric heterochromatin spread to juxtaposed euchromatic genes and cause stochastic gene silencing ([***Gowen and Gay, 1934***], reviewed in [***Girton and Johansen, 2008***; ***Elgin and Reuter, 2013***]). The extent of PEV is influenced by several genetic factors, including the amount of heterochromatic DNA in a genome (reviewed in [***Girton and Johansen, 2008***]) and heterochromatic enzymatic and structural proteins whose hypomorphic or null mutations enhance or suppress PEV (known as E(var)s and Su(var)s respectively, [***Elgin and Reuter, 2013***; ***Swenson et al., 2016***]). Likewise, the epigenetic effects of TEs on an adjacent reporter genes were observed to be contingent on the expression of two *Su(var)* genes (***Sentmanat and Elgin, 2012***).

Previously, we used *D. melanogaster* modEncode epigenomic data (***Nègre et al., 2011***) and demonstrated that histone H3K9me3, a key repressive epigenetic mark, is enriched around euchromatic TEs (***Lee 2015***). Importantly, TEs adjacent to *genes* that are highly enriched with H3K9me3 are more strongly selected against, supporting an important role for TE's epigenetic effects in its own population dynamics (***Hollister and Gaut, 2009***; ***Lee and Langley 2010***). Yet, several critical questions remain. For example, our previous study was based on the reference *D. melanogaster* strain (***Adams et al., 2000***), which has been maintained as a laboratory stock for many years, and is unlikely to be representative of natural populations. More importantly, single-strain analysis precluded distinguishing whether the enrichment of H3K9me3 at genes was due to TE-induced enrichment of repressive epigenetic marks, or the preferential insertions of TEs into genomic regions already enriched in repressive marks (***Lee 2015***).

To test the hypothesis that euchromatic TE insertions nucleate repressive epigenetic marks, here we exploit natural variation in the presence/absence of individual TE insertions in *D. melanogaster* populations. In this species, euchromatic insertions from most TE families segregate at low population frequencies (***Charlesworth and Langley, 1989***; ***Kofler et al., 2012***, ***2015***; ***Cridland et al., 2013***). Accordingly, randomly selected, unrelated individuals usually share few TE insertions. This will provide a direct comparison of epigenetic states at homologous sequences with and without the presence of TEs, and allow distinguishing the causal relationship between the presence of TEs and the enrichment of repressive epigenetic marks. Importantly, TEs only comprise 5.4% of the *D. melanogaster* euchromatic genome (***Hoskins et al., 2015***), and the epigenetic effects of *individual* TE insertions can thus be determined.

In this study, analyses of the epigenomes of two recently established, wild-derived, inbred *D. melanogaster* strains showed that euchromatic TEs are responsible for the enrichment of repressive epigenetic marks in flanking regions. Further, analysis of individual insertions revealed that more than half of euchromatic TEs are associated with epigenetic effects on flanking sequences, demonstrating their pervasive impact on the *Drosophila* genome. Importantly, we found evidence supporting stronger selection against TE insertions with more extensive epigenetic effects. Comparisons between the closely related *D. melanogaster* and *D. simulans* revealed that the epigenetic effects of TEs also vary between species, and correlate with variation in host genetic factors that regulate epigenetic silencing. Our results support that the epigenetic effects of euchromatic TEs, and host genetic factors that modulate these effects, play an important role in the population dynamics of TEs within and between species.

## Results

### Euchromatic TEs exhibit extensive epigenetic effects on adjacent sequences

In *Drosophila,* repressive histone modifications H3K9me2/3 (*Kouzarides, 2007*; *Grewal and Elgin, 2007*) and their cognate 'reader' protein Heterochromatin Protein 1a (HP1a) (*Eissenberg and Elgin, 2014*) play a dominant role in the initiation and maintenance of repressive chromatin states in heterochromatin. Our previous study showed that euchromatic sequences flanking TEs have strong enrichment for H3K9me3 in the reference *D. melanogaster* strain (*Lee 2015*). Using modEncode ChIP-seq data generated from the Oregon-R strain, we observed that sequences flanking euchromatic TEs are also enriched for another key heterochromatic histone modification (H3K9me2) and HP1a, while depleted for 'active' histone modifications H3K4me2 and H3K4me3, which are enriched at transcribing promoters (*Kouzarides, 2007*; *Kharchenko et al., 2011*) (*Figure 1*). Interestingly, enrichment for repressive epigenetic marks around TEs is strongest at the embryonic stage and weaker at later developmental stages, consistent with our previous study of only H3K9me3 (*Lee 2015*).

To investigate if the enrichment of repressive epigenetic marks is TE-induced or results from the preferential insertion of TEs into regions already enriched for repressive epigenetic marks, we performed Chromatin Immuno-Precipitation and sequencing (ChIP-seq) on H3K9me2 using two inbred,

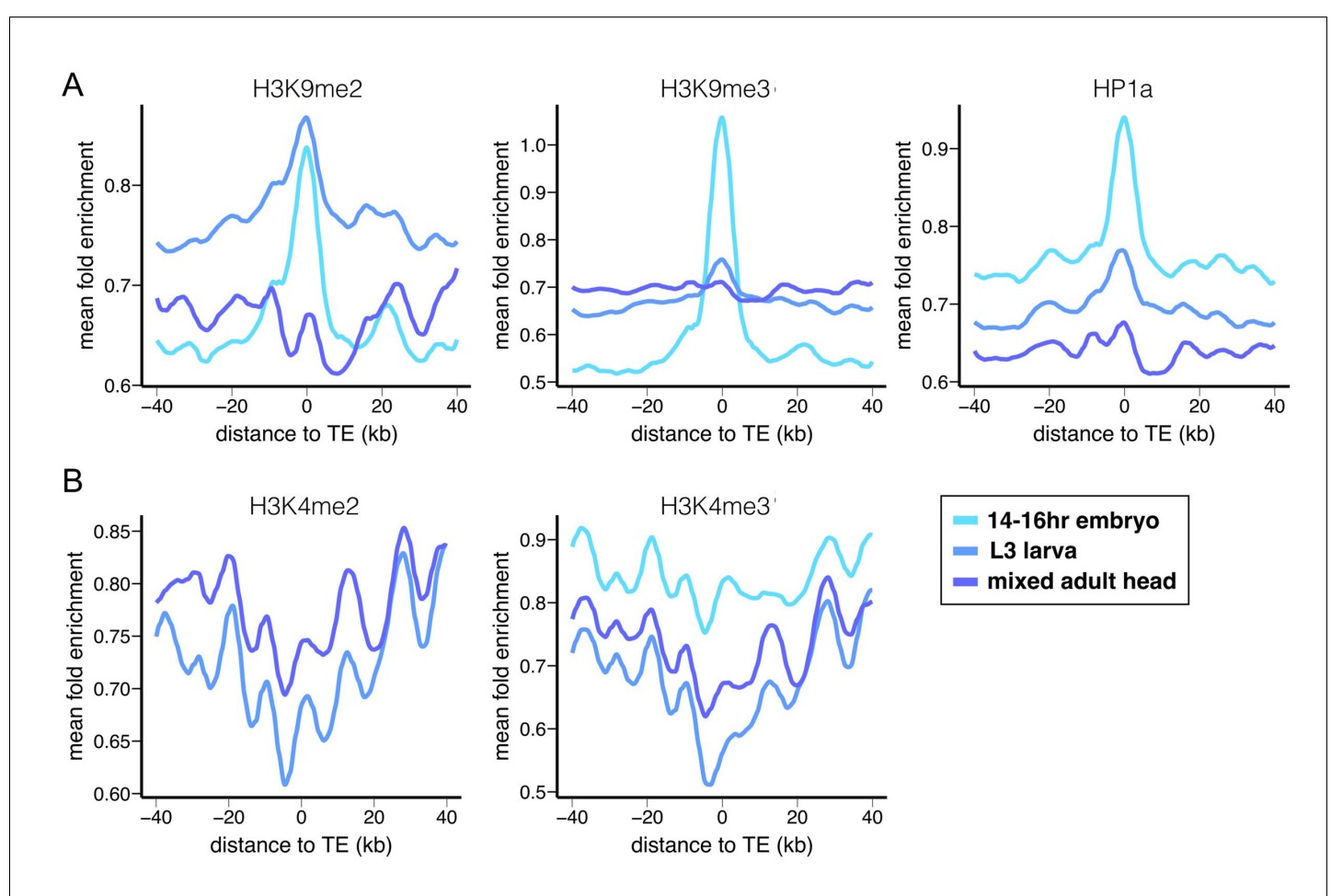

**Figure 1.** Epigenetic states of euchromatic sequences around TEs. Euchromatic sequences around TEs are enriched for (**A**) repressive epigenetic marks (H3K9me2, H3K9me3, and HP1a), (**B**) and depleted for active epigenetic marks (H3K4me2 and H3K4me2) in Oregon-R. Different colors represent different developmental stages. Plots were generated using LOESS smoothing (span = 10%).

wildtype *D. melanogaster* strains collected in North Carolina, USA (RAL315 and RAL360 from Drosophila Genetic Reference Panel or DGRP [*Mackay et al., 2012*]). These strains have been fully sequenced and annotated for the locations of euchromatic TE insertions (*Rahman et al., 2015*), allowing direct comparison of the epigenetic status of allelic regions with and without TEs. Our ChIP-Seq analyses of H3K9me2 distributions (see Materials and methods) in 4–8 hr RAL315 and RAL360 embryos, which contain fully-formed heterochromatin (*Yuan and O'Farrell, 2016*), only included TE insertions annotated with high confidence and unique to either strain (see Materials and methods). Importantly, we used highly conservative heterochromatin-euchromatin boundaries (0.5 Mb distal from epigenetically defined boundaries [*Riddle et al., 2011*]). This ensures that only euchromatic sequences and TEs were included in the analysis, and prevents confounding effects from pericentromeric or subtelomeric heterochromatin. Because the ChIP-Seq data were generated using whole animals that contain multiple cell types, combined with the stochastic nature of heterochromatic silencing, H3K9me2 enrichment reflects the *average* epigenetic states of all cells in the samples. Accordingly, we analyzed the enrichment of H3K9me2 quantitatively, instead of as binary states (see Materials and methods).

We compared the epigenetic states of euchromatic sequences around *all* TE insertions present in one strain with those of homologous alleles lacking the TE insertions in the other strain. The presence of TEs correlated with substantially higher H3K9me2 enrichment (*Figure 2*), which strongly supports the conclusion that these repressive mark enrichments are due to TE insertions, and not pre-existing epigenetic states. To quantify the epigenetic effects of individual TEs, we compared H3K9me2 fold enrichment in strains with and without a TE using non-overlapping 1 kb windows around each TE insertion (*Figure 2—figure supplement 1*). A TE was counted as having epigenetic effects if the H3K9me2 enrichment level was significantly higher in the strain with the TE than the other strain in the 0–1 kb windows flanking the TE insertion. We also estimated the 'extent of H3K9me2 spread' from the TE insertion (the farthest window in which H3K9me2 enrichment was consecutively and significantly higher in the strain with the TE) and the '% increase in H3K9me2 enrichment' (the difference in H3K9me2 enrichment between the two strains in 0–1 kb windows; see Materials and methods and *Figure 2—figure supplement 1*).

Surprisingly, more than half of the euchromatic TEs (54.2% of 419 TEs analyzed) were associated with enrichment for H3K9me2 in at least 1 kb of adjacent sequences. The TE-induced spreading of H3K9me2 in flanking sequence extended for a mean of 4.50 kb (standard deviation 4.59 kb), and an average of 79.8% increase in H3K9me2 enrichment at the TE insertion site (standard deviation 78.8%, *Figure 2—source data 1*). These observations revealed that the epigenetic effects of euchromatic TEs in *D. melanogaster* are not only pervasive and extensive, but also highly variable between TE insertions.

Previous investigations using randomly inserted transgenic constructs in *D. melanogaster* found that the epigenetic effects of TEs depend on proximity to pericentromeric or subtelomeric heterochromatin (*Sentmanat and Elgin, 2012*), and on local repeat density (*Huisinga et al., 2016*). Our analysis focused on regions far from these heterochromatic regions, and showed that TEs associated with H3K9me2 enrichment at flanking sequences are not concentrated around pericentromeric or subtelomeric heterochromatin (*Figure 2—figure supplement 2*). Also, local repeat density does not differ between TEs that are or are not associated with H3K9me2 spreading (*Mann-Whitney U test,* p=0.55). Similarly, we observed no correlations between the extent or magnitude of TE's epigenetic effects and local repeat density (*Spearman rank correlation test,* p=0.81 (repeat density vs extent of H3K9me2 spread), 0.65 (repeat density vs % increase in H3K9me2)). These results demonstrate that epigenetic influences of TEs are not restricted to specific genomic locations or contexts, and can be observed across diverse euchromatic regions.

## TE families of LTR-type and targeted by piRNAs show stronger epigenetic effects

While our results demonstrate that euchromatic TEs have widespread epigenetic effects in *D. melanogaster*, we also found that the epigenetic effects of individual TE insertions vary significantly. In particular, there is substantial variation in the epigenetic effects of insertions from different TE families (*Figure 3*). Many biological properties differ between TE families, including transposition mechanism (*Wicker et al., 2007*), genome abundance (*Kaminker et al., 2002*; *Quesneville et al., 2005*), and targeting by small RNAs (*Gunawardane et al., 2007*; *Brennecke et al., 2007*,

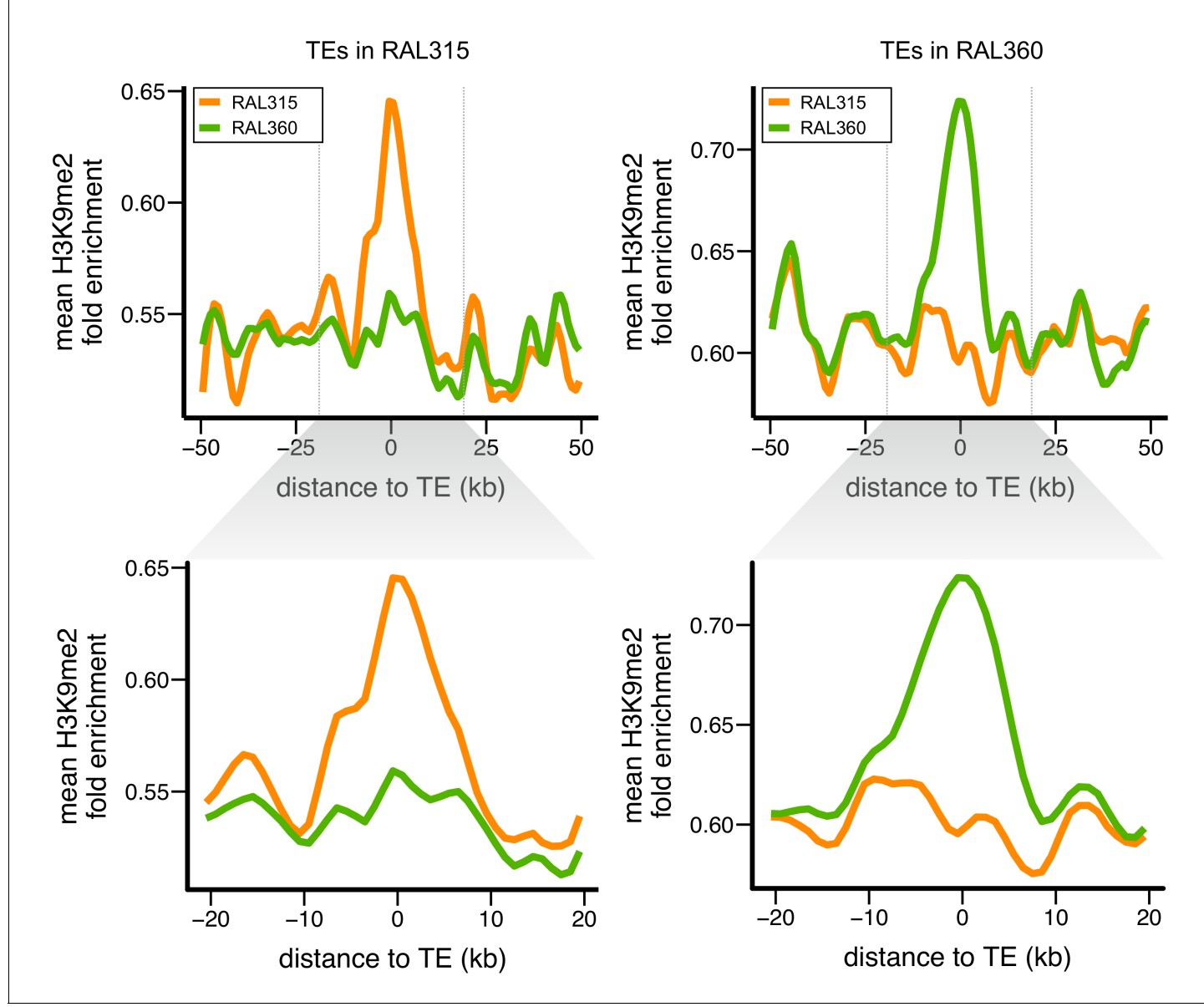

**Figure 2.** Euchromatic sequences around TE insertions are enriched for H3K9me2. Levels of H3K9me2 enrichment were compared between homologous sequences of two *D. melanogaster* strains. Left: sequences around TEs in strain RAL315 that are absent in RAL360. Right: sequences around TEs in strain RAL360 that are absent in RAL315. H3K9me2 fold enrichment was averaged over all euchromatic sequences flanking the analyzed TEs. Plots were generated using LOESS smoothing (span = 10%). Upper figures show ±50 kb around TE insertions, while lower figures show expanded views of ±20 kb.

The following source data and figure supplements are available for figure 2:

**Source data 1.** Estimates of epigenetic effects for *D. melanogaster* TE insertions.

**Figure supplement 1.** Three indexes describing the epigenetic effects of TE insertions.

**Figure supplement 2.** The distribution of TE insertions with epigenetic effects.

**Figure supplement 3.** IDR (irreproducible rate) analysis plots (*Li et al., 2011*) for replicates of *D. melanogaster* RAL strain ChIP samples.

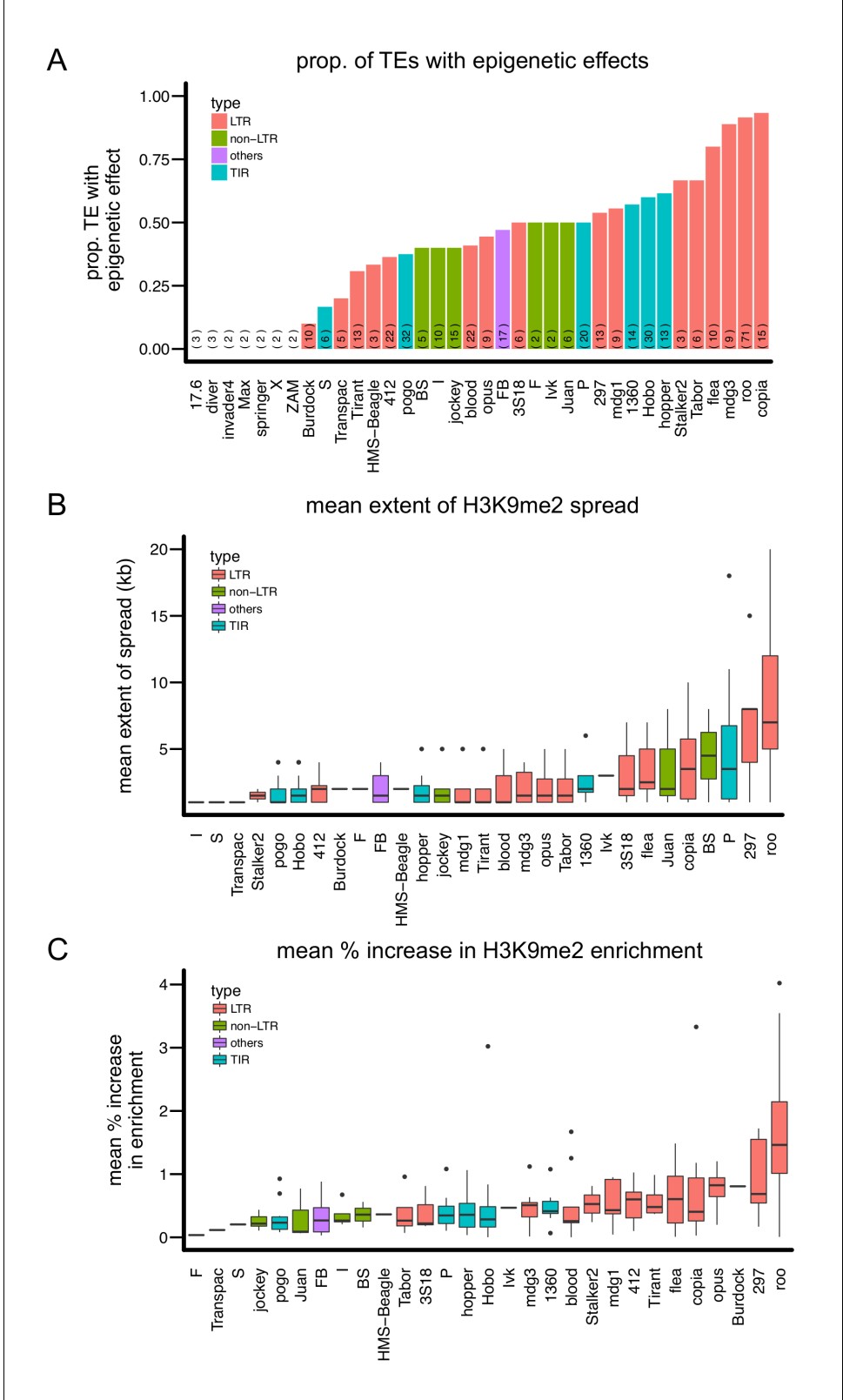

**Figure 3.** Variation in the epigenetic effects of different TE families. There is substantial variation in the (**A**) proportion of TEs with epigenetic effects, (**B**) mean extent of H3K9me2 spread, and (**C**) mean % increase in H3K9me2 enrichment of the TE families analyzed. Different colors denote different types of TEs. The number of observations for each TE family is in parenthesis in (**A**).

*2008*; *Ghildiyal et al., 2008*; *Czech et al., 2008*). We investigated which properties are associated with stronger epigenetic effects of insertions from a TE family.

Based on transposition mechanisms, there are three major types of TE families: Long Terminal Repeats (LTR) retrotransposons, non-LTR retroposons, and Terminal Inverted Repeats (TIR) transposons. An immediately obvious pattern is that LTR-type TE families seem to have the strongest epigenetic effects. The LTR *copia* family has the largest proportion of insertions with epigenetic effects, and LTR *roo* insertions display both the most extensive average spread of H3K9me2 and the largest average increase in H3K9me2 enrichment in flanking sequences (*Figure 3*). Similarly, eight of 11 TE families in which over half of analyzed insertions showed epigenetic effects are LTR-type, while the remaining three are TIR-type. The two TE families with >5 kb average spread of H3K9me2 and the four families that yield >50% mean increase in H3K9me2 enrichment are all LTR-type families. To formally test if LTR-type TE families have stronger epigenetic effects than other types of TE families, we estimated the *proportion* of TEs with epigenetic effects, the *average* extent of H3K9me2 spread, and *average* % increase in H3K9me2 enrichment of TE insertions from each TE family and compared these metrics between LTR-type and other types of TE families. Indeed, LTR-type TE families show a larger increase in H3K9me2 enrichment compared to other types of TEs (*Mann-Whitney U test* p=0.00047, median: 0.547 (LTR) vs 0.352 (others), *Figure 4*). The other two indexes are not significantly different, likely due to the high heterogeneity between LTR-type TE families (*Figure 4*).

In *Drosophila*, TEs are targeted by two types of small RNAs: piRNAs in the germline (*Gunawardane et al., 2007*; *Brennecke et al., 2007*) and endo-siRNAs in the soma (*Ghildiyal et al., 2008*; *Czech et al., 2008*). The epigenetic silencing of TEs in the germline and early embryo, which is maintained through development (*Gu and Elgin, 2013*), depends on piRNAs (*Klenov et al., 2007*; *Sentmanat and Elgin, 2012*; *Le Thomas et al., 2013*), while the role of endo-siRNAs in epigenetic silencing of TEs is currently less clear. Consistently, we observed that TE families targeted by more piRNAs show more extensive H3K9me2 spreading and enrichment in flanking sequences (*Table 1*). It is worth noting that there is no difference in the amount of piRNAs targeting LTR-type TEs compared to other major types of TEs (*Mann-Whitney U test, p*=0.19 (*w*K) and 0.39 (*w1118*)), suggesting that the observed correlation between the amount of piRNAs and TE's epigenetic effects was unlikely solely driven by stronger epigenetic effects of LTR-type TE families. On the other hand, we

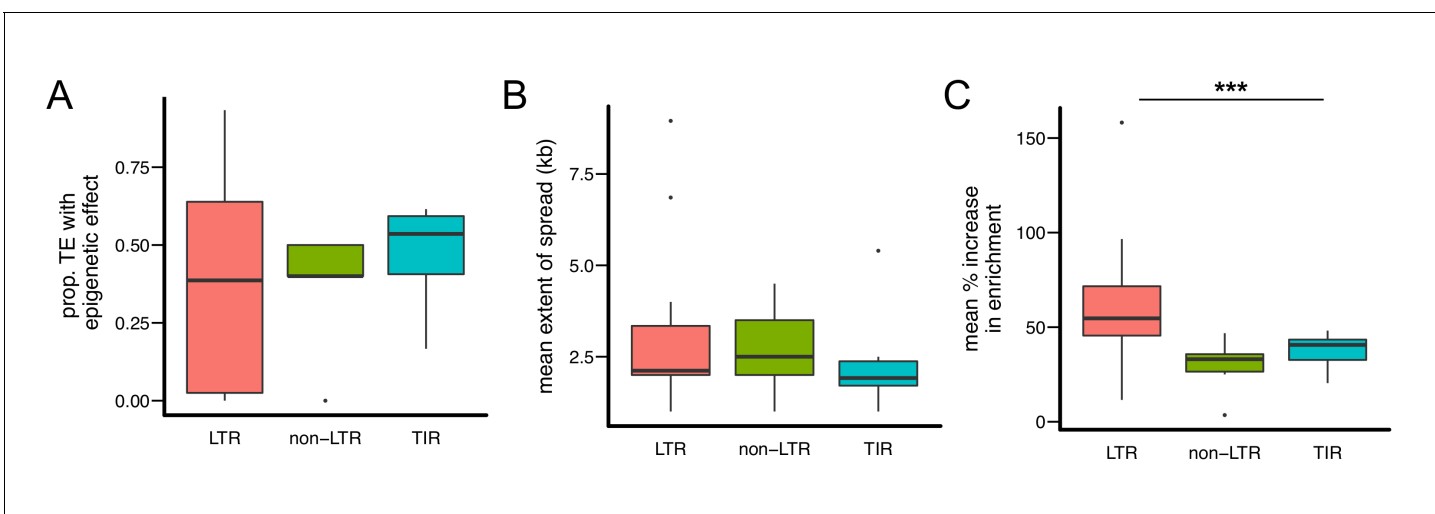

**Figure 4.** Quantitative analysis of the epigenetic effects of different types of TE families. While there are no significant differences in (**A**) the proportion of TEs with epigenetic effects and (**B**) the mean extent of H3K9me2 spread, (**C**) TE insertions of LTR-type families lead to significantly higher mean % increase of H3K9me2 enrichment in flanking sequences. Note that each data point represents one TE family. (*** Kruskal-Wallis test p<0.005).

The following figure supplement is available for figure 4:

**Figure supplement 1.** Scatter plot for the abundance of a TE family (X-axis) and the proportion of TEs with epigenetic effects (Y-axis).

**Table 1.** *Spearman rank correlation tests* between properties of TE families and the epigenetic effects of TEs. piRNA amounts were estimated from two studies (two genotypes: *w1118* and *wK*) and siRNA counts were estimated from two studies (**Ghildiyal et al., 2008**; **Czech et al., 2008**).

| | prop. TE with epigenetic effects | | mean extent of H3K9me2 spread | | mean % of increase in H3K9me2 enrichment | |
|---|---|---|---|---|---|---|
| | *p-value* | ρ | *p-value* | ρ | *p-value* | ρ |
| piRNA amount (w1118) | 5.18E-01 | 0.121 | **1.67E-02** | 0.465 | **1.13E-02** | 0.493 |
| piRNA amount (wK) | 9.99E-01 | 0.000 | **3.41E-03** | 0.553 | **7.09E-03** | 0.521 |
| siRNA counts (*Czech et al., 2008*) | 2.90E-01 | 0.193 | 4.99E-01 | 0.142 | 1.24E-01 | 0.316 |
| siRNA counts (*Ghildiyal et al., 2008*) | 6.08E-01 | 0.108 | 7.46E-01 | −0.075 | 1.46E-01 | 0.329 |
| family copy no. | **3.61E-03** | 0.473 | 6.24E-01 | 0.095 | 6.59E-01 | 0.085 |

did not find significant associations between the epigenetic effects of TEs and targeting by endo-siRNAs (*Table 1*).

It has been observed that insertions of abundant TE families are under stronger purifying selection than those of less abundant TE families, and several mechanisms were proposed to account for this copy-number dependency (reviewed in [*Barrón et al., 2014*]). Because the generation of piRNAs involves TE transcripts (*Gunawardane et al., 2007*; *Brennecke et al., 2007*), it was predicted that for a given TE family the epigenetic effects of TEs, and the associated strength of selection that influences the population dynamics of TEs, should also depend on TE copy number (*Lee and Langley 2010*; *Lee 2015*). Supporting this prediction, TE families with higher copy numbers in a large sample of African flies (*Kofler et al., 2015*) have larger proportions of insertions with epigenetic effects (*Table 1*). This strong correlation is not driven by TE families with exceptional abundance (*Figure 4— figure supplement 1*), because the removal of those TE families does not qualitatively change the results (*Spearman rank* ρ = 0. 46, p=0.0058). In summary, TE families of LTR-type, targeted by larger amounts of piRNAs, or of higher abundance display stronger epigenetic effects on adjacent sequences than other TE families.

## TEs with epigenetic effects are more strongly selected against

Given the high density of genes and other functional elements in *Drosophila* (*modENCODE Consortium et al., 2010*), H3K9me2 spreading from TEs to adjacent sequences is expected to have functional consequences. Accordingly, TE insertions with epigenetic effects should more likely be selected against and have lower population frequencies than TEs without H3K9me2 spreading.

Population genomic analysis indicated that Zambia is the likely ancestral origin of *D. melanogaster*, and Zambian populations have limited admixture from non-African genomes (*Pool et al., 2012*; *Lack et al., 2015*). Demographic history should thus have less effect on the analysis of TE population frequencies in the Zambian population compared to non-ancestral populations. Accordingly, we used genome sequences of a Zambian *D. melanogaster* population (*Lack et al., 2015*) to determine the population frequencies of individual TE insertions in the two DGRP strains analyzed (RAL315 and RAL360), which were first collected in North America. Consistent with previous genome-wide observations that most TE insertions have low population frequencies in *D. melanogaster* (*González et al., 2008*; *Kofler et al., 2012*, *2015*; *Cridland et al., 2013*), only 31.5% of TE insertions present in either of the two DGRP strains analyzed were found in the Zambian population, and these TEs displayed very low population frequencies (0.54% (first quartile), 0.56% (median), 1.61% (third quartile), *Figure 5—figure supplement 1*). We categorized TE insertions in the two DGRP strains according to their presence in the Zambian population ('high frequency' – present, 'low frequency' – not present). Low frequency TEs were more likely to exhibit spreading of H3K9me2 (*Fisher's Exact Test,* p=0.039, odds ratio = 1.58, *Figure 5A*), led to more extensive spreading (*Mann-Whitney U test,* p=0.011, *Figure 5B* and *Figure 5—figure supplement 2A*), and resulted in a larger increase in H3K9me2 enrichment (*Mann-Whitney U test,* p=0.014, *Figure 5C* and *Figure 5—figure supplement 2B*). Consistently, by analyzing the population frequencies of

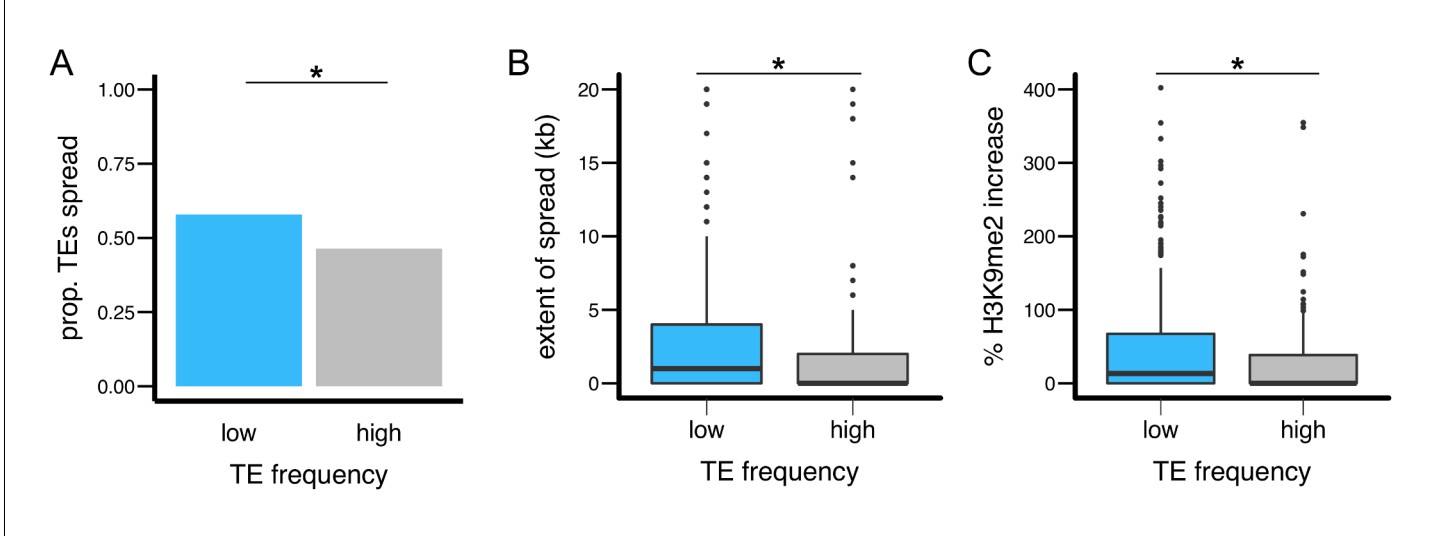

**Figure 5.** TEs with different population frequencies show different strength of epigenetic effects. TEs with low population frequencies are (**A**) more likely to show spread of H3K9me2, (**B**) result in more extensive spread of H3K9me2, and (**C**) lead to a larger increase in H3K9me2 enrichment. (*Mann-Whitney U test, p<0.05).

The following figure supplements are available for figure 5:

**Figure supplement 1.** Histogram for the population frequencies of analyzed TE insertions in the Zambian population.

**Figure supplement 2.** The epigenetic effects of TEs with low and high population frequencies.

**Figure supplement 3.** X-Y plot for TE's epigenetic effects and population frequencies in the Zambian population.

individual TE insertions, we observed significant negative correlations between the strength of a TE's epigenetic effects and its population frequency (*Spearman rank* $\rho$ = −0.15 (extent of H3K9me2 spread) and −0.14 increase in H3K9me2), p<0.005 for both, *Figure 5—figure supplement 3*).

A potential confounding factor for our observation is that the population frequencies of TE insertions vary between TE families (i.e. insertions from specific TE families tend to have high/low population frequencies, [*Petrov et al., 2011*; *Kofler et al., 2012*, *2015*]). Thus, 'low' and 'high' frequency categories of TE insertions could be comprised of insertions from different TE families, whose variation in population frequencies could be due to factors other than the differential strength of selection removing TE insertions (*Blumenstiel, 2011*; *Blumenstiel et al., 2014*). To address this issue, we performed multiple regression analyses that jointly consider the impact of TE's epigenetic effects and family identity on the population frequencies of TEs (see Materials and methods). Because most TEs in the two DGRP strains analyzed were not detected in the Zambian population (*Figure 5—figure supplement 1*), we treated the frequency of TE insertions (the number of individuals in which a TE insertion is present in the Zambian population) also as dichotomous variable ('high frequency' TE or not, see Materials and methods). Even accounting for the effect of TE family identity, the regression coefficients for TE's epigenetic effects on population frequencies are still negative for all the regression models analyzed, and are statistically significant for a majority of the models (*Table 2*), suggesting that TE family identity is unlikely a major contributor for the negative associations between TE's epigenetic effects and population frequencies.

An alternative explanation for the observed negative associations between TE's epigenetic effects and population frequencies is that TEs *without* epigenetic effects tend to occur in regions of low meiotic recombination. TE insertions in regions with low meiotic recombination are repeatedly observed to have higher population frequencies than TEs in other genomic regions (*Charlesworth and Lapid, 1989*; *Charlesworth et al., 1992*; *Bartolomé and Maside, 2004*; *Kofler et al., 2012*; *Cridland et al., 2013*). A lower probability of recombination between TE

**Table 2.** Regression analysis for the associations between TE's epigenetic effects and population frequencies while accounting for the influence of TE family identity. Population frequencies of individual TE insertion (response variable) were modeled as either dichotomous variable ('high frequency' TE or not) or count (TE count). Because the distribution of TE count is overdispersed, TE count was modeled as either 'quasipoisson' or 'negative binomial' in regression analyses. The influence of TE family identity was treated as either fixed or random effect. Also see **Table 2—source data 1** for regression coefficients for all TE families.

| Response variable | Family identity | Extent of spread | | Magnitude of spread | |
|---|---|---|---|---|---|
| | | *p-value* | **Regression coefficient** | *p-value* | **Regression coefficient** |
| 'high frequency' TE or not | fixed effect | 4.72E-01 | −0.029 | 3.37E-01 | −0.246 |
| | random effect | 1.58E-01 | −0.049 | 4.83E-02 | −0.409 |
| TE count (quasipoisson) | fixed effect | 4.00E-03 | −0.188 | 4.73E-03 | −1.121 |
| | random effect | 3.20E-03 | −0.136 | 1.71E-04 | −1.400 |
| TE count (negative binomial) | fixed effect | 5.25E-04 | −0.151 | 2.31E-04 | −1.041 |
| | random effect | 9.19E-05 | −0.138 | 5.49E-05 | −0.986 |

**Source data 1.** Regression coefficients for the epigenetic effects of TEs (extent of spread and magnitude of spread) and each TE family.

insertions at different genomic locations (ectopic exchange [*Langley et al., 1988*; *Montgomery et al., 1991*]) and/or reduced efficacy of selection against TEs due to selective interference (*Hill and Robertson, 1966*; *Felsenstein, 1974*) have been proposed to account for these observations (reviewed in [*Charlesworth and Langley, 1989*; *Lee and Langley 2010*; *Barrón et al., 2014*]). However, we observed that recombination rates do not differ between TEs with or without spreading of H3K9me2 (*Mann-Whitney U test,* p=0.83). Similarly, neither the extent of H3K9me2 spread nor the increase in H3K9me2 enrichment in flanking sequences was correlated with the local recombination rate for individual TE insertions (*Spearman rank correlation test,* p=0.62 (recombination rate vs extent of H3K9me2 spread) and 0.55 (recombination rate vs % increase in H3K9me2 enrichment)). It is unlikely that variation in recombination rates can account for the observations that TEs with stronger epigenetic effects have lower population frequencies. Overall, these results strongly support the proposed selection against the epigenetic effects of TEs.

## Epigenetic effects of TEs result in differential epigenetic states of adjacent coding genes

We hypothesized that selection against TEs with epigenetic effects result from the associated functional consequences, in particular influences on the epigenetic states of adjacent functional elements. To investigate the predicted epigenetic influence of TEs on adjacent genes, we categorized *euchromatic* protein coding genes according to their shortest distance to a TE (0–1 kb, 1–2 kb, 2–5 kb, 5–10 kb, and no TE within 10 kb; see Materials and methods). Within each of the two strains analyzed, genes in proximity to TEs are more enriched for H3K9me2 (*Figure 6—figure supplement 1*), consistent with previous observations (*Lee 2015*).

To investigate the influence of TEs on the epigenetic states of homologous alleles, we calculated a z-score that compares the H3K9me2 enrichment of genic alleles with and without TEs located within 10 kb (see Materials and methods). The absolute value of the z-score reports the magnitude of differences in H3K9me2 enrichment between homologous alleles in the two strains, and the sign indicates if the allele with adjacent TEs has higher H3K9me2 enrichment (positive: yes, negative: no). As expected, genes with adjacent TEs in either strain have significantly higher, positive z-scores compared to genes distant from TEs in both strains (*Figure 6A*). We further investigated if the differential epigenetic states between homologous alleles depend on TE-induced epigenetic effects, or only on the presence of TEs. For all categories of genes within 10 kb from TEs, z-scores are significantly higher for genes whose neighboring TEs exhibit H3K9me2 spreading (*Figure 6B*). Consistently, there are significant positive correlations between the z-scores of genes and the extent of epigenetic effects from the nearest TEs (vs. the extent of H3K9me2 spread: *Spearman rank* $\rho$ = 0.31, p<$10^{-15}$; vs. % increase in H3K9me2 enrichment: *Spearman rank* $\rho$ = 0.30, p<$10^{-15}$). It is worth noting that

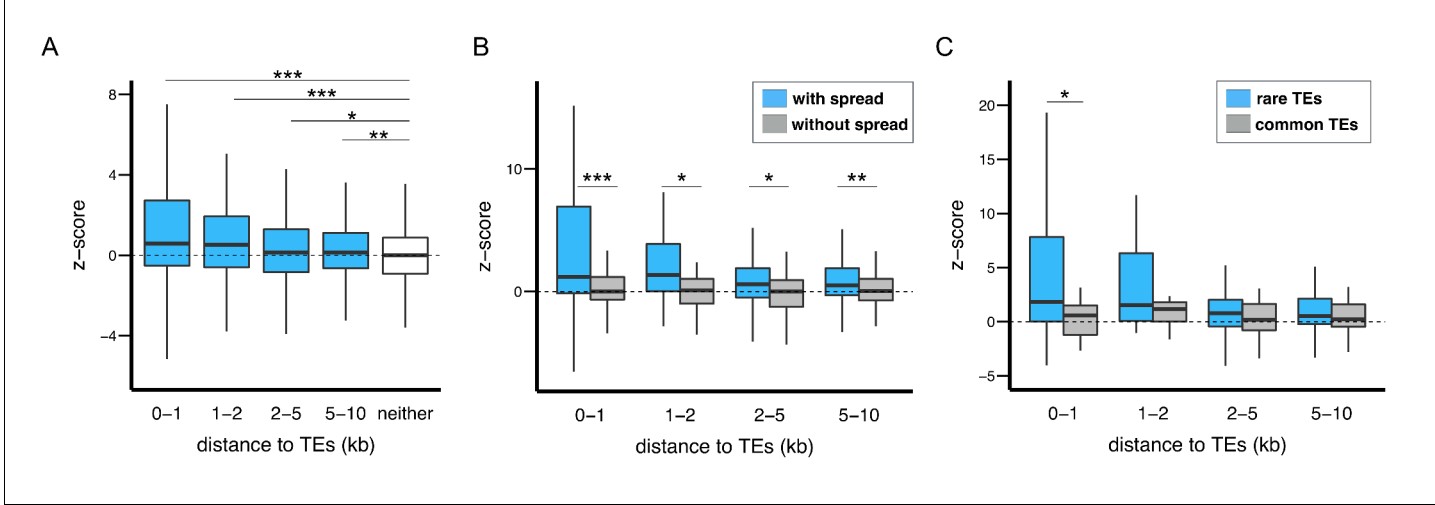

**Figure 6.** The epigenetic effects of TEs on adjacent protein coding genes. (**A**) Alleles with adjacent TEs have higher H3K9me2 enrichment compared to homologous alleles in the strain that lacks adjacent TEs, as indicated by positive z-scores (see text), and the strength of the effect decreases with distance from TEs. (**B**) Genes adjacent to TEs with epigenetic effects show stronger differential enrichment for H3K9me2 than genes adjacent to TEs without epigenetic effects. (**C**) Genes adjacent to low frequency TEs with epigenetic effects, which likely experienced stronger selection against them, show stronger differential enrichment of H3K9me2 than genes adjacent to high frequency TEs with epigenetic effects (*Mann-Whitney U test*, *p<0.05, **p<0.01, ***p<0.001).
The following figure supplement is available for figure 6:

**Figure supplement 1.** H3K9me2 enrichments at genes decreases with distance from TEs in both RAL315 (left) and RAL360 (right).

genes whose nearest TEs did not exhibit epigenetic effects have similar z-scores to genes without TEs within 10 kb (dashed line, *Figure 6B*). These observations demonstrate that the spread of repressive epigenetic marks from euchromatic TEs leads to substantial epigenetic differences at homologous alleles of adjacent coding genes.

TE insertions with stronger enrichment of repressive marks in adjacent functional alleles should lead to more deleterious functional consequences. We predict these TEs should be under stronger purifying selection and have lower population frequencies than other TEs. To test this hypothesis, we further restricted the analysis to genes whose nearest TEs show spreading of H3K9me2. Among these genes, those whose nearest TEs were absent in the Zambian population ('low frequency' TEs, see above) have significantly higher z-scores than genes adjacent to 'high frequency' TEs (*Mann-Whitney U test*, p=0.0019, median: 0.95 (genes near low frequency TEs) vs 0.32 (genes near high frequency TEs)). The observed differences are most prominent for genes within 1 kb of TEs (*Figure 6C*). Consistently, there is a significant negative correlation between the z-score of a gene and the population frequency of its nearest TE (*Spearman rank* $\rho = -0.18$, p<10$^{-3}$).

A potential functional consequence of H3K9me2 enrichment is reduced transcript levels of influenced alleles. To address this possibility, we performed RNA-seq of developmental stage-matched embryos. Within either strain, there are indeed significant negative correlations between H3K9me2 enrichment and transcript levels for genes within 10 kb of TEs (*Spearman rank* $\rho = -0.35$ (RAL315) and $-0.33$ (RAL360), p<10$^{-16}$ for both). To compare the differential epigenetic states and transcript levels between homologous alleles, we calculated fold changes in expression and z-scores for H3K9me2 enrichment level using RAL 360 allele as reference (*Figure 7A*). Note this is different from the z-score used above, which uses the allele without TE as reference. We found an excess number of genes that support the influence of TE's epigenetic effects on gene expression (higher H3K9me2 enrichment and lower expression of alleles with adjacent TEs, shaded green area in *Figure 7A*) for genes with TEs within 10 kb when compared to other genes in the genome (*Figure 7B*, upper 2 × 2 Tables). Restricting the analysis to TEs with epigenetic effects produced an even larger proportion of genes whose TE-neighboring alleles have higher H3K9me2 enrichment and lower expression

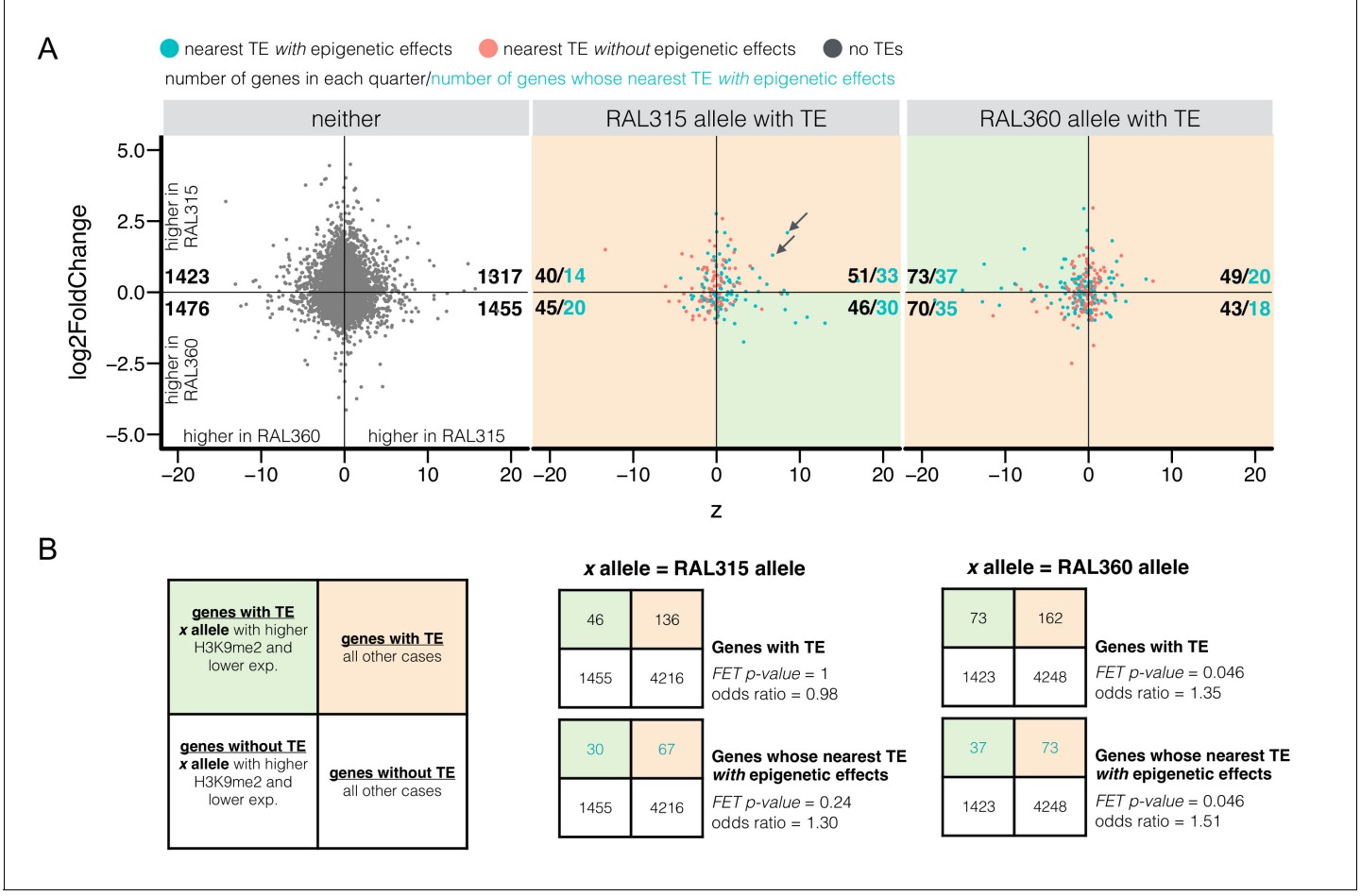

**Figure 7.** Differential H3K9me2 enrichment and RNA transcript levels of protein coding genes with and without adjacent TE insertions. (**A**) The z-score for H3K9me2 enrichment (X-axis) and log2 expression fold change (Y-axis) were plotted for euchromatic protein coding genes without TEs within 10 kb ('neither') and for genes with adjacent TEs in either strain. It is worth noting that both H3K9me2 z-score and log2 expression fold change used RAL360 as reference. Shaded green areas are genes displaying the expected negative influence of TE's epigenetic effects on gene expression (i.e. alleles adjacent to TEs have higher H3K9me2 enrichment and lower RNA transcript levels), while shaded orange areas are all other cases of epigenetic states and transcript levels. For each sub-plot, the numbers of genes (blue, pink, or gray dots) in each quarter are shown in black, and the numbers of genes whose nearest TEs *with* epigenetic effects (blue dots) are shown in blue. (**B**) Left: 2 × 2 contingency table for comparing the number of genes supporting the influence of TE's epigenetic effects on gene expression (shaded green) and the number of other genes (shaded orange), against those for genes without TEs within 10 kb ('neither'). Middle and right: 2 × 2 contingency tables for testing if there is an excess number of genes with TEs in RAL315 (middle) and in RAL360 (right) supporting the influence of TE-induced epigenetic effects on gene expression.

(*Figure 7B*, bottom 2 × 2 Tables). Intriguingly, the excess number of genes supporting TE-induced epigenetic effects on expression was mainly observed for one of the two strains analyzed (RAL360). Furthermore, while we found a weak, but significant, negative correlation between z-scores for H3K9me2 enrichment and fold changes in expression for genes without TEs (*Spearman rank* $\rho = -0.035$, p=0.0084), there are no such correlations observed for genes with TEs within 10 kb (*Spearman rank test* p=0. 57 (RAL315) and 0.16 (RAL360)). In fact, there are multiple genes whose alleles associated with TEs have higher enrichment of H3K9me2, but also higher expression (e.g. arrows in *Figure 7A*). These observations suggest that the influence of TE-induced epigenetic states on gene expression may be more complex (see Discussion).

## Stronger epigenetic effects of TEs in *D. simulans* compared to *D. melanogaster*

*D. simulans* diverged from *D. melanogaster* only four million years ago (*Obbard et al., 2012*), yet is widely observed to harbor fewer TE insertions compared to *D. melanogaster* (*Dowsett and Young, 1982*; *Vieira et al., 1999*; *Vieira and Biémont, 2004*; *Kofler et al., 2015*). We hypothesized that variation in the epigenetic effects of TEs, and thus strength of selection against them, contributes to this between-species difference in TE content. To test this hypothesis, we performed H3K9me2 ChIP-seq on 4–8 hr embryos from the *D. simulans* reference strain. Similar to *D. melanogaster*, we observed strong H3K9me2 enrichment in sequences flanking TEs (*Figure 8A*), and genes adjacent to TEs have higher H3K9me2 enrichment than genes distant from TEs (*Figure 8—figure supplement 1*). Furthermore, genes adjacent to TEs with epigenetic effects (see below) have higher H3K9me2 enrichment than genes adjacent to TEs without epigenetic effects (*Figure 8—figure supplement 1*). For genes within 10 kb of TEs, there is also a strong negative correlation between H3K9me2 enrichment and transcript levels (*Spearman rank $\rho$ = −0.45, p<10$^{-16}$*), supporting a functional consequence of H3K9me2 enrichment in *D. simulans*.

To compare the epigenetic effects of TEs in *D. melanogaster* and *D. simulans,* the H3K9me2 enrichment at sequences flanking TEs were estimated relative to the median fold enrichment level at sequences 20–40 kb away from TEs in both species (see Materials and methods). Current TE annotations are *D. melanogaster*-centric and it is likely that our analysis missed TE families and/or variants that are *D. simulans*-specific. Accordingly, we restricted comparisons to TE families that have at least two insertions in both species. While there are no significant between-species differences in the extent of H3K9me2 spread (*paired MWU test,* p=0.67), TE families in *D. simulans* display a larger proportion of TEs with epigenetic effects (*paired MWU test,* p=0.013), and a larger increase in relative H3K9me2 fold enrichment (*paired MWU test*, p=0.035; *Figure 8B*). It is worth noting that only high confidence TE calls were included in the analysis, and only 14 families had at least two copies in both species. The *roo* family has the highest proportion of TEs with epigenetic effects in both species (94.4% and 86.5% in *D. melanogaster* and *D. simulans*, respectively), while the *1360* family has the largest differences between the two species (14.3% and 77.8% in *D. melanogaster* and *D. simulans*, respectively). These results demonstrate that TEs in *D. simulans* exhibit stronger epigenetic effects on flanking sequences compared to *D. melanogaster.*

## Variation in genetic modifiers of PEV correlates with differences in the epigenetic effects of TEs between species

The extent of heterochromatin-mediated gene silencing (e.g. PEV) depends on several genetic modifiers, in particular the *amount* of heterochromatic DNA in a genome (reviewed in [*Girton and Johansen, 2008*]), and the *dosage* of several *Su(var)* and *E(var)* genes, whose wildtype proteins enhance and weaken PEV respectively (*Elgin and Reuter, 2013*; *Swenson et al., 2016*). The prevailing model is that altering the ratio of heterochromatin targets (heterochromatic DNAs) and regulators (Su(var) and E(var) proteins) influences heterochromatin nucleation and spreading (*Locke et al., 1988*). For example, lower amounts of heterochromatic DNA result in increased levels of heterochromatic Su(var) proteins in other regions, and accordingly enhance PEV. Because both PEV and the epigenetic effects of TEs are mediated through spreading of the same repressive epigenetic marks (H3K9me2/3 and HP1a), the epigenetic effects of TEs may depend on similar PEV modifiers. Indeed, a limited survey using reporter constructs demonstrated that the epigenetic effects of TEs depend on the expression of HP1a (Su(var)205) and Su(var)3–9, which binds and catalytically generates H3K9me2/3 marks, respectively (*Sentmanat and Elgin, 2012*). We thus predicted that variation in the epigenetic effects of TEs within and between species, and accordingly genomic abundance of TE insertions, could be due to differences in the amounts of heterochromatic DNA and/or modifier proteins.

We investigated the hypothesis that stronger epigenetic effects of TEs in *D. simulans* are associated with lower amounts of heterochromatic DNA or altered expression of *Su(var)s/E(var)*s. In *Drosophila,* heterochromatic DNA consists of simple repeats and degenerate TEs (*Hoskins et al., 2007, 2015*). We first identified short repeats (12-mers) that are enriched in heterochromatic regions by performing K-mer analysis of H3K9me2 ChIP-Seq data (see above), and then quantified the amount of identified H3K9me2-enriched 12-mers in these two species using published genomic sequencing data ((*Kofler et al., 2015*), see Materials and methods). Consistent with previous quantitation of

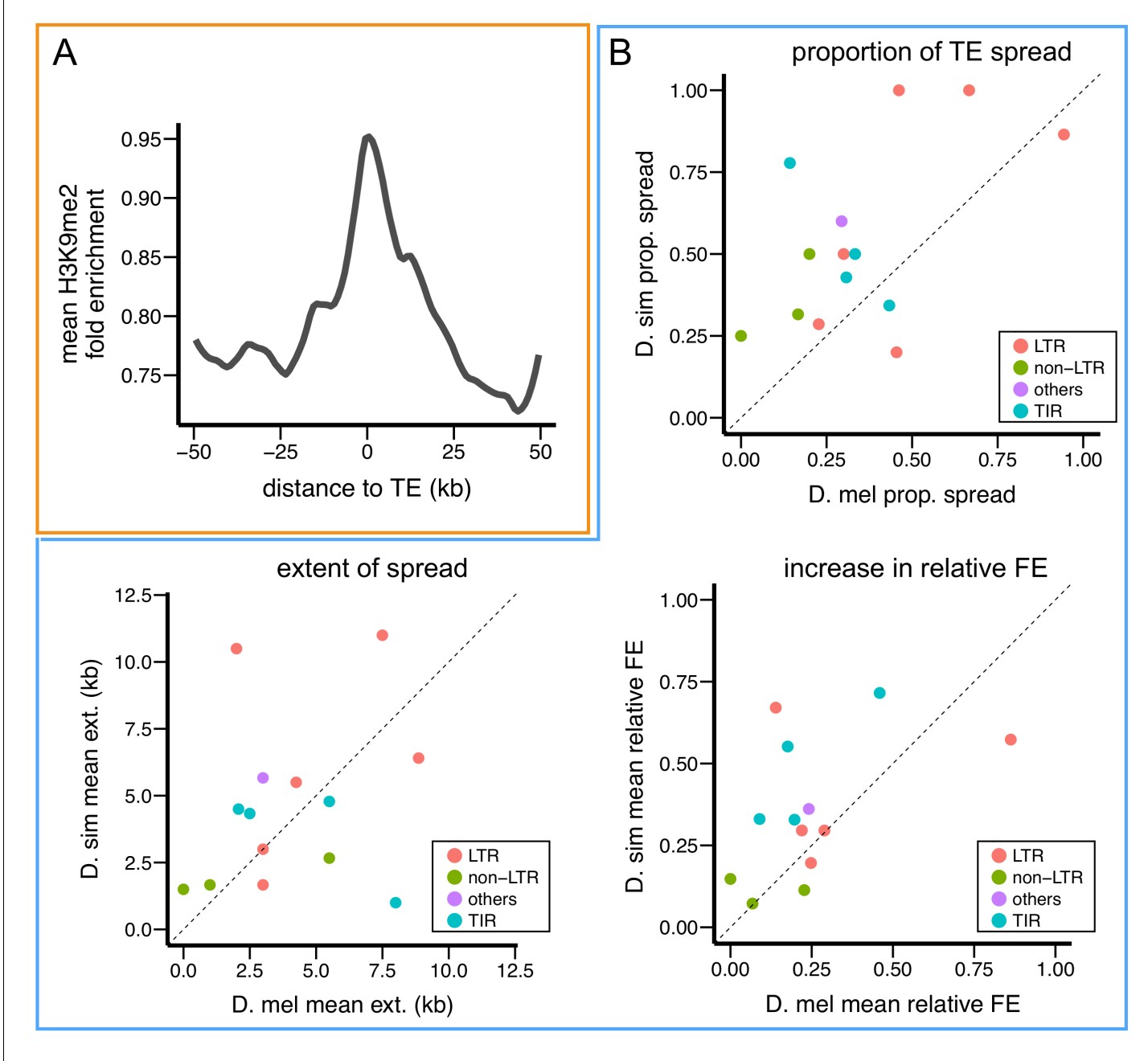

**Figure 8.** *D. simulans* TEs show stronger epigenetic effects than *D. melanogaster* TEs. (**A**) Enrichment of H3K9me2 is also observed at sequences adjacent to euchromatic TEs in *D. simulans*. (**B**) Compared to insertions of the same TE family in *D. melanogaster,* insertions in *D. simulans* are more likely to show epigenetic effects (proportion of TE spread) and a larger increase in relative H3K9me2 fold enrichment in adjacent sequences. FE: fold enrichment, D. mel: *D. melanogaster,* D. sim: *D. simulans*.

The following source data and figure supplements are available for figure 8:

**Source data 1.** Estimates of epigenetic effects for *D. simulans* TEs.

**Figure supplement 1.** H3K9me2 enrichment at genes adjacent to TEs in *D. simulans*.

**Figure supplement 2.** IDR plots for replicates of *D. simulans* ChIP samples.

simple repeat content using orthogonal approaches (melting curves [*Lohe and Brutlag, 1987*] or flow cytometry [*Bosco et al., 2007*]), we observed lower amounts of H3K9me2-enriched simple repeats in *D. simulans* compared to *D. melanogaster* (*Figure 9A*, *ANOVA* p-value=0.00013 (species) and $<10^{-6}$ (library preparation method)).

To test if the expression of *Su(var)s* and *E(var)s* varies between the two species, we estimated z-scores for between-species differences in expression rank (high expression – low rank), using *D. simulans* as reference (i.e. positive z-socre: higher expression in *D. simulans*; negative z-score: higher expression in *D. melanogaster*). Compared to other genes in the genome, 40 known *Su(var)s*, as a group, have higher expression in *D. simulans* than in *D. melanogaster* (*Kolmogorov–Smirnov test*, p$<10^{-6}$, *Figure 9B*; also see *Figure 9—figure supplement 1* and *Figure 9—source*

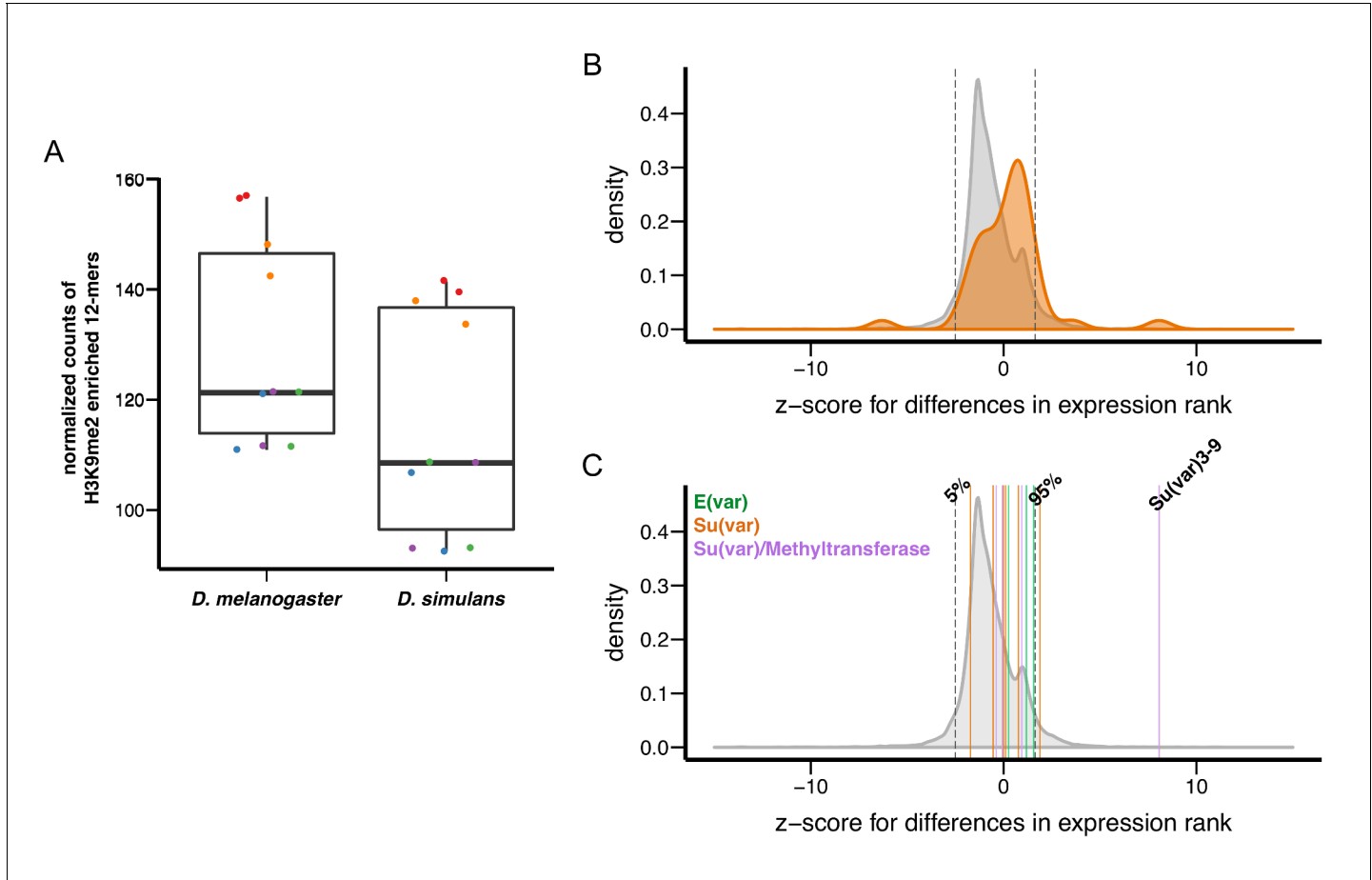

**Figure 9.** Variation in genetic modifiers of PEV in *D. melanogaster* and *D. simulans*. (A) *D. simulans* has higher normalized amounts of H3K9me2-enriched 12-mers than *D. melanogaster*. Raw amounts of H3K9me2-enriched 12-mers were normalized with sequencing coverage in each sample before comparisons (see Materials and methods). Different library preparation methods (see [*Kofler et al., 2015*]) are denoted with dots of different colors. (B) Compared to genome-wide distributions (shaded gray), known Su(var) genes as a group (orange, 40 genes in total) have higher expression in *D. simulans* than in *D. melanogaster*. Positive z-score represents lower expression rank (i.e. higher expression) in *D. simulans* than in *D. melanogaster*. Dashed vertical lines represent the top and bottom 5% of transcript level differences genome-wide. (C) Z-score for differences in transcript levels of ten known dosage-dependent E(var) genes (green), Su(var) genes (orange), and histone methyltransferase genes (also Su(var)s) between *D. melanogaster* and *D. simulans* are denoted as vertical lines and compared to genome-wide distributions (shaded gray). Dashed vertical lines indicate top and bottom 5% of transcript level differences genome-wide.

The following source data and figure supplement are available for figure 9:

**Source data 1.** Gene expression level of *Su(var)* and *E(var)* genes in *D. melanogaster* and *D. simulans*.

**Figure supplement 1.** Differences in transcript levels of *Su(var)* and *E(var)* genes between *D. melanogaster* and *D. simulans*.

*data 1* for individual genes included in the analysis). The small number of known *E(var)s* (five) precluded us from drawing any solid conclusions (*Figure 9—figure supplement 1*). Several *Su(var)s/E(var)s* are known to have *dosage-dependent* effects on heterochromatin silencing (*Elgin and Reuter, 2013*; *Swenson et al., 2016*). Among these *dosage-dependent Su(var)s/E(var)s*, *Su(var)3–9* showed significantly higher expression in *D. simulans* than in *D. melanogaster* (*Figure 9C*). Overall, we found that *D. simulans* has lower amounts of H3K9me2-enriched simple repeats and higher expression of *Su(var)s* compared to *D. melanogaster*, both of which could account for the stronger epigenetic effects of TEs observed in *D. simulans*.

## Discussion

Despite the presence of TEs in virtually all eukaryotic genomes surveyed, there is wide variation in euchromatic TE content among species, demonstrated by significant differences in copy number (*Clark et al., 2007*; *Biémont, 2010*; *Chalopin et al., 2015*), frequency spectra (*Lockton and Gaut, 2010*; *Agren et al., 2014*; *Kofler et al., 2015*), predominant types of TEs (*Vieira and Biémont, 2004*; *Chalopin et al., 2015*; *Kofler et al., 2015*), and species-specific TE families (*Daniels et al., 1990*; *Lohe et al., 1995*; *Mills et al., 2006*). Understanding the causes for such variation is critical for evaluating the impacts of various evolutionary forces on the population dynamics of TEs. Selection against the deleterious effects of TEs has been theoretically proposed (*Charlesworth and Charlesworth, 1983*) and empirically supported (reviewed in [*Charlesworth and Langley, 1989*; *Lee and Langley 2010*; *Barrón et al., 2014*]) as a dominant force restricting the selfish increase of TEs, and shaping variation of TEs within and between species. Differences in effective population size ([*Lynch and Conery, 2003*; *Lockton et al., 2008*], but see [*Charlesworth and Barton, 2004*; *Groth and Blumenstiel, 2017*]), mating systems (*Charlesworth and Charlesworth, 1995*; *Wright and Schoen, 1999*; *Dolgin et al., 2008*; *Agren et al., 2014*), and modes of reproduction (e.g. asexual vs sexual, [*Arkhipova and Meselson, 2000*]), were suggested to influence the efficacy of selection against TEs, and thus result in divergence in euchromatic TE content between species. In addition, differential exposure to opportunities for horizontal transfer might also contribute to variation in TE profiles (*Clark et al., 2007*; *Groth and Blumenstiel, 2017*).

In this study, we analyzed the *epigenetic* influence of TEs and investigated the role of such effects in the population dynamics of TEs. By comparing the epigenome of two *D. melanogaster* strains with divergent TE insertion positions, we conclude that the enrichment of repressive epigenetic marks around euchromatic TEs is due to the presence of TEs, instead of the preferential insertions of TEs into genomic regions already enriched for repressive epigenetic marks. Surprisingly, quantification of the epigenetic effects of individual TE insertions in *D. melanogaster* revealed that more than half of the euchromatic TEs analyzed are associated with at least 1 kb spread of repressive epigenetic marks, with an average spread of 4.5 kb from TEs that display epigenetic effects. In contrast, repressive DNA methylation from epigenetically silenced TEs in *A. thaliana* predominantly spreads only a few hundred base pairs (*Quadrana et al., 2016*; *Stuart et al., 2016*). Since ~20% of the euchromatic genes are within 5 kb from at least one TE insertion in the reference *D. melanogaster* strain (*Kaminker et al., 2002*; *Quesneville et al., 2005*; *Hoskins et al., 2015*), these estimates suggest that the epigenetic influence of TEs on functional sequences is extensive in *D. melanogaster*. Indeed, we observed strong positive associations between the presence of TEs with epigenetic effects and the enrichment of repressive epigenetic marks in adjacent genic alleles, when compared to homologous alleles lacking a neighboring TE insertion. This concurs with observations in *Arabidopsis* that TEs contribute significantly to genic DMRs (differential methylated regions) between genomes (*Schmitz et al., 2013*; *Quadrana et al., 2016*; *Stuart et al., 2016*). In addition, we found substantial variation in the epigenetic effects among TE families, which could be due to differences in TE types, targeting by piRNAs, and/or the abundance of TE families.

It was proposed that heterochromatin assembly depends on the concentration of essential heterochromatic enzymatic and structural proteins, whose concentration is highest in heterochromatin and decreases with increasing distance ('mass action model', [*Locke et al., 1988*]). This model can explain the spread of repressive epigenetic marks from pericentromeric or subtelomeric heterochromatin to juxtaposed euchromatic genes in PEV (reviewed in [*Girton and Johansen, 2008*; *Elgin and Reuter, 2013*]), and likely the enrichment of repressive epigenetic marks at sequences adjacent to euchromatic TEs. Consistent with predictions of this model, previous (*Lee 2015*) and current

analyses both found that H3K9me2/3 enrichment decays with distance from TEs. The enrichment of repressive epigenetic marks could also result from TE-induced formation of de novo piRNA-generating loci that include TE-flanking sequences (*Shpiz et al., 2014*). piRNAs from these piRNA-generating loci would accordingly initiate epigenetic silencing of TE-flanking euchromatic sequences.

TE-induced reduction in expression of neighboring genes has been demonstrated in *Drosophila* (*Cridland et al., 2015*; *Lee 2015*), and the spreading of repressive epigenetic marks from TEs is a plausible mechanism for such observations. We observed an excess of TE-flanking alleles with higher H3K9me2 enrichment and lower transcript levels than homologous alleles lacking nearby TE insertions. However, there are apparent exceptions to this pattern, such as alleles adjacent to TEs having higher enrichment of H3K9me2 but also higher expression levels than homologous alleles lacking neighboring TEs (e.g. arrows in *Figure 7A*). In addition to the epigenetic effects of TEs, *cis*-regulatory sequences contained within TEs could interfere with gene regulation, leading to increased as well as reduced gene expression (*Naito et al., 2009*; *Batut et al., 2013*). In fact, recent studies that jointly analyzed mobilomes and transcriptomes in *A. thaliana* populations found that TE insertions result in equal frequencies of increased and decreased expression of flanking genes (*Quadrana et al., 2016*; *Stuart et al., 2016*). Besides, SNPs and copy number variants (CNVs) in regulatory sequences are also identified as significant contributors to differential expression between homologous alleles (*Massouras et al., 2012*). The observed variation in transcripts level is thus the joint consequence of TE's epigenetic effects and other genetic factors. Importantly, there are known exceptions to the association between higher enrichment of heterochromatic marks and lower gene expression. In fact, enrichment of repressive epigenetic marks and heterochromatin proteins is surprisingly high for active genes normally located in the pericentromeric heterochromatin, and is required for the proper expression of these genes (*Wakimoto and Hearn, 1990*; *Hearn et al., 1991*; *Yasuhara and Wakimoto, 2008*; *Riddle et al., 2011*). For some of these genes, the association with a local heterochromatin environment was even observed for homologs located in euchromatic regions of other *Drosophila* species (*Caizzi et al., 2016*).

Importantly, our quantifications of the epigenetic effects of TEs and the associated functional consequences on gene expression are likely underestimates of the real effects in natural populations. Screens for genetic mutations repeatedly report an inverse correlation between the dominance of a deleterious allele and its fitness effect (reviewed in [*Simmons and Crow, 1977*; *Wilkie, 1994*; *Osada et al., 2009*]). If similar trends apply to the deleterious epigenetic effects of TEs, establishment of the wild-derived inbred *Drosophila* strains used here would have removed the majority of TEs with lethal or sub-lethal epigenetic effects (i.e. substantial functional consequences). The small sample size used here (two strains) precluded detecting the subtle functional consequences of TE-induced epigenetic effects that are expected to be prevalent in inbred strains. Future larger scale epigenomic and transcriptomic profiling of multiple, diverse population samples would be necessary to further investigate the functional consequence of TE's epigenetic effects. It is also worth noting that not every gene included in the analysis is expressed at the embryonic stage studied (4–8 hr embryo). In fact, we observed a deficiency in the number of genes that both have adjacent TEs and are expressed in 4–8 hr embryos (using data from modEncode (*Graveley et al., 2011*); *Fisher's Exact Test,* p=0.00017, odds ratio = 0.73). This deficiency is even stronger when only considering TEs with epigenetic effects (*Fisher's Exact Test,* $p<10^{-5}$, odds ratio = 0.58). Epigenetic marks established at embryonic stages are expected to influence both somatic and germline cells, and were experimentally demonstrated to have a long lasting functional effect through development (*Gu and Elgin, 2013*). Accordingly, our observations are consistent with selection preferentially removing TEs that result in spreading of repressive epigenetic marks to adjacent genes expressed in embryonic stages.

Compared to neutral TE insertions that confer no fitness effects, TEs exerting deleterious fitness effects are more strongly selected against and should appear rare in populations (reviewed in [*Charlesworth and Langley, 1989*; *Lee and Langley 2010*; *Barrón et al., 2014*]). Consistent with the prediction that the epigenetic effects of TEs have deleterious fitness consequences, we found that TE insertions with stronger epigenetic effects have lower population frequencies, demonstrating the importance of such effects in the population dynamics of TEs. In addition, theoretical work suggests that stable equilibrium of TE copy number in an outbreeding, meiotically recombining population requires synergistic epistasis of the deleterious effects of TEs; specifically host fitness must decrease faster than linear with respect to increases in TE copy number (*Charlesworth and Charlesworth, 1983*). However, despite being critical to explaining the population dynamics of TEs,

synergistic epistasis has only been empirically supported for deleterious effects mediated by ectopic recombination between nonhomologous TE insertions (*Langley et al., 1988*; *Petrov et al., 2003*, *2011*), one of the many proposed deleterious *genetic* mechanisms for TEs (reviewed in [*Lee and Langley 2010*]). Because piRNAs are generated through feed-forward cycles that involve TE transcripts (*Gunawardane et al., 2007*; *Brennecke et al., 2007*), we previously predicted that the deleterious epigenetic effects of TEs would confer the theoretically required synergistic epistasis for stable containment of TEs (*Lee and Langley 2010*; *Lee 2015*). The observed association between the abundance of a TE family and the propensity of its members to influence the epigenetic states of adjacent sequences supports this prediction, and further extends possible evolutionary mechanisms for stable containment of TE copy number in host populations.

An especially interesting observation is the stronger epigenetic effects of TEs in *D. simulans* compared to *D. melanogaster*. TE insertions in *D. simulans* are more likely to show spreading of H3K9me2 and result in larger increase in H3K9me2 enrichment compared to those of the same TE family in *D. melanogaster*. All else being equal, the stronger epigenetic effects should lead to stronger selection removing TE insertions in *D. simulans.* If the rate of TE proliferation is also similar in these two species, this could account for the lower genomic TE content (*Clark et al., 2007*) and fewer TE insertions (*Dowsett and Young, 1982*; *Vieira et al., 1999*; *Vieira and Biémont, 2004*; *Kofler et al., 2015*) in *D. simulans* compared to *D. melanogaster.* Our observations complement previous comparisons of *A. thaliana* and *A. lyrata*, which revealed negative associations between genomic TE content and the effectiveness of siRNA targeting (*Hollister et al., 2011*). Overall, these results strongly support the conclusion that variation in the epigenetic effects of TEs contributes to the divergent TE content observed between even closely related species and significantly impacts TE evolution.

Finally, we attempted to gain insights into the molecular mechanisms that could account for the between-species differences in the epigenetic effects of TEs. We observed lower amounts of heterochromatic DNA and higher expression of *Su(var)* genes in *D. simulans*, both of which are known to generate more extensive spread of repressive epigenetic marks from constitutive heterochromatin in *D. melanogaster*. The amount of heterochromatic DNA was among the first identified dosage-dependent PEV modifiers ([*Dimitri and Pisano, 1989*], reviewed in [*Girton and Johansen, 2008*]). Similarly, *Su(var)* genes were identified through mutants that *suppress* PEV, demonstrating that the wild type genes play positive roles in heterochromatin establishment and/or maintenance (reviewed in [*Girton and Johansen, 2008*; *Elgin and Reuter, 2013*]). In particular, *Su(var)3–9*, which encodes the H3K9 methyltransferase, displays significantly higher expression in *D. simulans* than in *D. melanogaster*, and its between-species difference in expression level ranks in the top 0.75% genome-wide (see Results). H3K9 methylation is critical for suppression of TE expression and transposition (*Penke et al., 2016*), and *Su(var)3–9* mutations reduce the epigenetic effects of TEs on adjacent reporter genes (*Sentmanat and Elgin, 2012*). Furthermore, *Su(var) 3–9* is a haploid suppressor and triploid enhancer of PEV (*Schotta et al., 2002*), suggesting that changes in transcript levels of *Su (var)3–9* would result in quantitative differences in the epigenetic effects of TEs. Assuming that the epigenetic effects of TEs depend on the amount of heterochromatic DNA and the expression of *Su (var)* genes in *D. melanogaster* and *D. simulans* similarly, variation in these two genetic PEV modifiers provides a viable explanation for the observed differences in the epigenetic effects of TEs and more broadly divergent TE profiles between these two species.

Our observations support the hypothesis that the host genetic environment contributes to the extent of deleterious epigenetic effects of TEs and influences the population dynamics of TEs, pointing towards a rarely addressed mechanism for the widely observed variation of TEs. Furthermore, PEV is long known to be temperature sensitive (*Gowen and Gay, 1933*), and several abiotic factors influence heterochromatin function (*Seong et al., 2012*; *Silver-Morse and Li, 2013*). Thus, different environmental conditions present in diverse habitats could also contribute to variation in the epigenetic effects of TEs and the widely divergent TE profiles within and between species.

The observed significant variation in genetic PEV modifiers between *D. melanogaster* and *D. simulans* raises questions about its evolutionary causes. It is worth noting that the deleterious epigenetic effects of TEs are considered as a side effect of host-directed epigenetic silencing of TEs (*Hollister and Gaut, 2009*; *Lee 2015*), and direct positive selection for stronger epigenetic effects of TEs would be unlikely to explain the between-species differences in genetic PEV modifiers. Elevated transcript levels of *Su(var)* genes might have been selected for to silence burst expansions of

specific TE families or other types of repetitive sequences. Alternatively, *Su(var)* genes are highly pleiotropic (reviewed in [*Girton and Johansen, 2008*; *Eissenberg and Reuter, 2009*; *Elgin and Reuter, 2013*]), and selection might have acted instead on their essential chromosomal functions, with varying influence on the epigenetic consequences of TEs as a secondary effect. Similarly, changes in the amounts of heterochromatic DNA could have resulted from selfish expansion of repetitive sequences (*Charlesworth et al., 1994*) and/or global changes in chromatin landscapes due to karyotype turnover (*Kaiser and Bachtrog, 2010*; *Vicoso and Bachtrog, 2013*). Our findings suggest that the evolution of TEs may be more tightly associated with the evolution of other cellular, chromosomal, and/or genetic processes than previously appreciated.

## Materials and methods

### Drosophila strains

Drosophila strains used in this study are *D. melanogaster* RAL315 (Bloomington Drosophila stock center (BDSC) #25181), RAL360 (BDSC #25186), and *D. simulans* w501 (Drosophila species stock center). Previous analysis showed that these two *D. melanogaster* inbred wildtype strains have low residual heterozygosity (*Lack et al., 2015*). Flies were cultured on standard medium at 25°C, 12 hr light/12 hr dark cycles.

### ChiP-Seq and RNA-Seq experiments

Before collecting embryos, mated flies were allowed to lay eggs on fresh apple juice agar plates for one hour. Embryos were then collected on fresh apple juice agar plates for 4 hr and aged for 4 hr (to enrich for 4–8 hr embryos). All fly rearing and embryo collections were performed at 25°C. Chromatin isolation and immunoprecipitation were performed following the modEncode protocol (http://www.modencode.org/). The antibody used for H3K9me2 (abcam 1220) was validated by modEncode and showed high consistency between lots (*Egelhofer et al., 2011*). For each strain, there were at least two replicates and each IP replicate had a matching input. ChIP-Seq libraries were prepared with NuGen Ovation Ultralow Library Systems V2 (San Carlos, CA) and sequenced on Illumina Hi-Seq with 100 bp, paired-end reads. RNAs were extracted from embryos that were collected using the same procedures using the RNeasy Plus kit (Qiagen). There were two replicates for each strain. RNA-Seq libraries were prepared using Illumina TruSeq and sequenced on Illumina Hi-Seq with 100 bp, paired-end reads.

### TE calls

We used highly conservative euchromatin-heterochromatin boundaries: 0.5 Mb distal from those reported previously for *D. melanogaster* (*Riddle et al., 2011*). For *D. simulans,* we used boundaries that are 0.5 Mb distal from the sharp transition in H3K9me2 enrichment, based on our ChIP-Seq data. For all the analyses reported, we excluded TEs, genes, and sequences in heterochromatic regions. For *D. melanogaster* strains, we used TE insertions reported with strong confidence (coverage ratio greater than or equal to 3; [*Rahman et al., 2015*]). TEs that are shared between two RAL strains, in shared H3K9me2 peaks in euchromatin (called by MACS2 and present in both strains, see below), and/or in exons were also excluded. TEs in the *D. simulans* genome were annotated according to (*Chiu et al., 2013*), using blastn (*Camacho et al., 2009*). In brief, we used the blast hit with smallest e-value and excluded a putative insertion when the blast hit had the same smallest e-value for more than one TE family. We required a putative TE call to have at least 100 bp, at least 80% identity to canonical TEs, and merged TE calls of the same family and within 500 bp. TEs of different families but were within 2 kb were called as putative TE clusters and excluded from the analysis. In both species, we excluded INE-1 TEs, most which are relics of a TE family that experienced an ancient burst of transposition events and are now mostly fixed in populations (*Kapitonov and Jurka, 2003*; *Singh and Petrov, 2004*). Our study included 255 TEs for the Oregon-R strain, 419 TEs for RAL strains, and 349 TEs for the *D. simulans* strain.

### ChIP-Seq data analysis

Raw reads were processed with trim-galore ('Babraham Bioinformatics - Trim Galore!") to remove adaptors and low quality sequences. Processed reads were mapped to release six reference *D.*

*melanogaster* genome (*Hoskins et al., 2015*) or release two reference *D. simulans* genome (*Hu et al.* 2013), using bwa mem with default parameters (v 0.7.5) (*Li and Durbin, 2009*). Reads with mapping quality score lower than 30 were filtered using samtools (*Li, 2011*) and excluded from further analysis. We used Macs2 with a liberal significance threshold (p=0.2) to generate peak calls for IDR (irreproducible rate) analysis (*Li et al., 2011*), which evaluates the reproducibility of ChIP replicates. Replicates for our samples had low IDRs (*Figure 2—figure supplement 3* and *Figure 8—figure supplement 2*), and were combined to generate a single H3K9me2 fold enrichment track (between IP and matching input) for each sample. Our analyses were based these fold-enrichment tracks.

The baseline H3K9me2 enrichment level is slightly different between the two *D. melanogaster* strains, potentially due to technical and/or biological reasons. As the enrichment of repressive epigenetic marks is generally confined to 10 kb from TEs (*Lee 2015*), we used the H3K9me2 enrichment levels 20–40 kb upstream and downstream of each TE insertion to normalize the background levels between the two strains. For each annotated TE insertion, we divided its flanking 20 kb upstream and 20 kb downstream sequences into 20 nonoverlapping 1 kb windows respectively (*Figure 2—figure supplement 1*). We then used *Mann-Whitney U test* to assess if H3K9me2 enrichment in the $i^{th}$ upstream and downstream windows differs significantly between the two strains. The most distant windows considered are 20 kb from TE insertions. The 'extent of H3K9me2 spread' is the farthest windows in which the H3K9me2 enrichment is consecutively and significantly higher in the strain with TE. When the farthest windows are different between the left and right sides of a TE insertion, we used the window closer to TE for the 'extent of H3K9me2 spread' (to be conservative). The '% increase of H3K9me2' is the difference of median H3K9me2 enrichment between the two strains in the 0–1 kb windows immediately next to TEs (with TE strain minus without TE strain), divided by the enrichment level for the strain without TE.

For *D. simulans* TEs, we calculated *relative fold enrichment* with respect to the median H3K9me2 fold enrichment at flanking 20–40 kb upstream or downstream sequences, whichever had a higher median (to be conservative). *D. melanogaster* data were also analyzed using this method to allow between-species comparisons. We again used *Mann-Whitney U test* to assess if the relative H3K9me2 enrichment in a window is significantly higher than one, the background level of relative fold enrichment. Here, the 'extent of spread' is the farthest window in which the relative fold enrichment is consecutively and significantly higher than one. The 'increase in fold enrichment' is the median relative fold enrichment in the 0–1 kb window immediately next to TE, minus one. To evaluate the performance of this method, we compared *D. melanogaster* results using this method to those based on normalization between strains. We found significant correlations between the two approaches for indexes of TE's epigenetic effects (*Spearman rank* $\rho$ = 0.63 (extent of H3K9me2 spread) and 0.68 (increase in H3K9me2 enrichment), p<$10^{-16}$ for both). The calls for the presence of epigenetic effects (extent of spread at least 1 kb) were consistent between the two methods for 73.3% of TEs. Among TEs with inconsistent results, 67.8% (18.1% of all TEs) were called as 'no epigenetic effect' by the single-genome method but 'with epigenetic effect' by the method that incorporate both strains, suggesting that the single-genome method is overall more conservative in estimating the epigenetic effects of TEs. For 80% of the TEs, the estimated extents of spread were either the same or differ within 2 kb between the two methods.

We estimated the percentage of sites annotated as simple repeats in a 10 kb window around each TE insertion (based on the repeat-masked release 6 *D. melanogaster* genome from https://genome.ucsc.edu/). Recombination rate estimates for TE insertions were interpolated from (*Comeron et al., 2012*), which reported average recombination rate of *D. melanogaster* in 1 Mb window. For TE-family level analysis, we only considered TE families with at least two observations. Abundance of each TE family is based on (*Kofler et al., 2015*). Ovarian *piRNA* sequences for two wildtype strains (*w1118* and *wK*) were from (*Brennecke et al., 2008*; *Kelleher et al., 2012*), and the normalized count estimates of each TE family were from (*Kelleher and Barbash, 2013*). We used two endo-siRNA datasets: (1) the reported counts of endo-siRNA (excluded pre-microRNAs) in adult heads for each TE family (*Ghildiyal et al., 2008*), and (2) endo-siRNAs generated by Ago2 pull-down libraries from ovaries (*Czech and Hannon, 2011*). The raw endo-siRNA sequences were processed with trim-galore, mapped with bwa aln to all annotated TEs in *D. melanogaster* reference genome, and counted for each TE family. For both piRNAs (from [*Kelleher and Barbash, 2013*]) and siRNAs, those that mapped to more than one TE families were excluded from the analysis.

For gene-based analysis, we calculated average fold enrichment over gene bodies for each replicate and used quantile-normalization. We calculated a z-score for each gene (mean H3K9me2 enrichment of allele with nearby TE – mean H3K9me2 enrichment of allele lacking nearby TE)/(mean standard deviation of both strains). Our analysis excluded genes with ambiguous TE presence/absence status (such as a gene with TE within 2 kb in one strain but with TE within 5 kb in the other strain). We used *D. melanogaster* annotation 6.07 and *D. simulans* annotation 2.01.

ChIP-Seq data from Oregon-R were downloaded from the modEncode website (http://www.mod-encode.org/) and analyzed with the same procedures.

## RNA-seq analysis

Raw reads were processed with trim-galore, followed by mapping to release six reference *D. melanogaster* genome (*Hoskins et al., 2015*) or release two reference *D. simulans* genome (*Hu et al. 2013*) using TopHat with default parameters (*Trapnell et al., 2009*). We used htseq-count (*Anders et al., 2015*) to count the number of reads mapping to exons and used DESeq2 (*Love et al., 2014*) to normalize and estimate expressional fold change between the two *D. melanogaster* strains. Estimates of transcript abundance were highly correlated between biological replicates (*Pearson's r* = 0.98 (RAL315), 0.97 (RAL360), and 0.88 (*D. simulans*), $p<10^{-16}$ for all). We only analyzed genes annotated as expressed in 4–8 hr embryos by the modEncode developmental time course study (*Graveley et al., 2011*). Indeed, genes annotated as *no or extremely low* expression in 4–8 hr embryos in the modEncode study have much fewer mapped reads than other genes in our RNA-seq data (median for RPKM, RAL315: 0.058 (not expressed) vs 13.16 (other genes), RAL360: 0.031 (not expressed) vs 12.70 (other genes), *Mann-Whitney U test,* $p<10^{-16}$ for both). To investigate the functional consequence of TE-induced enrichment of repressive epigenetic marks, we categorized protein-coding genes according to their epigenetic states (RAL 315 or RAL360 higher?) and RNA transcript levels (RAL 315 or RAL360 higher?) in the two strains. The proportion of genes with predicted TE-induced epigenetic states and RNA transcript levels (higher H3K9me2 enrichment and lower expression for alleles adjacent to TEs) for genes with TEs in 10 kb were compared to other genes in the genome using *Fisher's Exact Test* (also see *Figure 7*).

To compare expression levels between the two species, we used RPKM (reads per kilobase per million reads) and ranked genes from highest (small rank) to lowest (large rank) expression in each library. Z-score was calculated as (mean rank of *D. melanogaster* – mean rank of *D. simulans*)/mean standard deviation. A negative z-score represents higher expression in *D. melanogaster* while a positive z-score represents higher expression in *D. simulans.*

## TE population frequency analysis

Raw reads from Drosophila Population Genomic Project (DPGP) 3 (*Lack et al., 2015*) were mapped to release six *D. melanogaster* reference genome using bwa mem with default parameters. Reads mapping within 500 bp upstream or downstream of TE insertion sites were parsed out using samtools. Parsed reads were assembled using phrap (*Ewing and Green, 1998*) following parameters in (*Cridland et al., 2013*). Assembled contigs were aligned against repeat-masked release six *D. melanogaster* genome using blastn. If one of the contigs spanned over at least 50 bp on both sides of a TE insertion site, the TE was called absent in the analyzed genome. If no contigs spanned the TE insertion site, contigs were aligned against canonical TE sequences and sequences of all TEs in the reference *D. melanogaster* genome using blastn. When there were blast hits to TE sequences, a TE was called present if there was a contig aligning at least 30 bp left or right of the TE insertion site without spanning the insertion site. All other scenarios were called as missing data. For population frequency analysis, we only included TEs that have at least 100 alleles (out of 197 alleles) called in DPGP3 genomes.

A large proportion of the analyzed TE insertions (68.5%) has zero population frequencies in the Zambian population (*Figure 5—figure supplement 1*). Accordingly, in some analyses, we also categorized TEs into those that are present in the Zambian population ('high frequency' TEs) and those that are absent ('low frequency' TEs). To account for the influence of TE family identity on TE's population frequencies, we performed regression analysis using generalized linear model and generalized mixed linear model. Population frequencies of TE insertions (response variable) were treated as either dichotomous variable ('high frequency' TE or not) or count (the number of individuals in which

**Table 3.** Comparisons of H3K9me2 enriched 12-mers between *D. melanogaster* and *D. simulans* using different normalization and thresholds. Raw counts of H3K9me2 enriched 12-mers were normalized by read coverage of either the orthologous exonic regions or all orthologous genomic regions. 'Fold enrichment threshold' is the threshold for a 12-mer to be considered as H3K9me2 enriched in the ChIP-Seq data. '% of 12-mers' is the proportion of H3K9me2 enriched 12-mers among all 12-mers.

| normalization | fold enrichment threshold | % of total 12-mers | ANOVA *p-value* Species | library preparation method |
|---|---|---|---|---|
| exon reads | 1.5 | 20.21% | 1.29E-04 | 1.43E-07 |
| exon reads | 2 | 12.89% | 8.62E-05 | 1.28E-07 |
| exon reads | 3 | 6.41% | 1.01E-02 | 1.55E-07 |
| all reads | 1.5 | 20.21% | 2.07E-09 | 2.40E-05 |
| all reads | 2 | 12.89% | 5.14E-11 | 1.70E-05 |
| all reads | 3 | 6.41% | 1.60E-03 | 1.20E-02 |

a TE insertion is present). Because the distribution of TE count is overdispersed (i.e. the variance is greater than the mean), we modeled the TE count as having either 'quasipoission' or 'negative binomial' distribution. The influence of TE family identity was modeled as either fixed effect (generalized linear model) or random effect (generalized mixed linear model). The two indexes for the epigenetic effects of TEs ('extent of H3K9me2 spread' and '% increase in H3K9me2 enrichment') were analyzed separately. Regression models used were:

$$logit\ p \sim TE's\ epigenetic\ effects\ (either\ ''extent\ of\ H3K9me2\ spread''\ or\ ''\%\ increase\ in\ H3K9me2\ enrichment'') + family$$

$$TE\ count \sim TE's\ epigenetic\ effects + family$$

where *logit p* is the log odds of whether a TE is observed in the Zambia population ('high frequency' TEs). We used MASS (**Venables and Ripley, 2002**) for negative binomial regression and lme4 (**Bates et al., 2015**) for generalized mixed linear models in R.

## Heterochromatic repeat content
### Identifying heterochromatic repeats
To identify repeats enriched in heterochromatic regions, we used KMC2 (**Deorowicz et al., 2015**) to quantify the number of 12-mers in IP and Input libraries from the H3K9me2 ChIP-seq experiments. To normalize between sequencing libraries, we divided the counts of 12-mers by the number of reads that mapped uniquely to the reference genome with at least 30 mapping quality score. The idea is to have a measure of the abundance of repetitive sequence relative to the single-copy regions of the genome. To find 12-mers enriched with H3K9me2, we divided the normalized counts from IP libraries by the normalized counts in corresponding Input libraries, and considered 12-mers with at least 1.5 fold enrichment with H3K9me2 as enriched in heterochromatic regions.

### Quantifying the amount of heterochromatic repeats
We used pooled-genome sequencing of *D. melanogaster* and *D. simulans* data from (**Kofler et al., 2015**) to quantify the amount of heterochromatic repeats in these two species. It is worth noting that in (**Kofler et al., 2015**), libraries of these two species were prepared and sequenced in pairs, which minimized technical variations. We used KMC2 with the same parameters to count the number of H3K9me2 enriched 12-mers in each library. To account for the different completeness of reference genomes for *D. melanogaster* and *D. simulans*, and the variation in sequencing coverage between samples, we counted the number of reads mapped uniquely and with at least 30 mapping quality score to Flybase annotated orthologous exonic regions. Numbers of H3K9me2 enriched 12-mers were then normalized with sequencing coverage in these orthologous exonic regions. Results

using different fold enrichment thresholds to identify H3K9me2 enriched 12-mers or using different normalization metrics gave consistent results (*Table 3*).

## Acknowledgements

We thank Julie Cridland, Robert Kofler, and Nelson Lau for discussions and data for the annotations of TE insertions, Sasha Langley and other members of the Karpen lab for helpful discussions of the project, and Hsiao-Han Chang for helpful discussions regarding statistical analysis. We greatly appreciate Andrea Betancourt and Mia Levine for critically reading earlier version of the manuscript, and Magnus Nordborg, Brandon Gaut, and one anonymous reviewer for their helpful and constructive comments. This study was supported by NIH R01 GM117420 to GHK.

## Additional information

### Funding

| Funder | Grant reference number | Author |
|---|---|---|
| National Institute of General Medical Sciences | R01 GM117420 | Gary H Karpen |

The funders had no role in study design, data collection and interpretation, or the decision to submit the work for publication.

### Author contributions

YCGL, Conceptualization, Data curation, Formal analysis, Investigation, Methodology, Writing—original draft, Writing—review and editing; GHK, Conceptualization, Resources, Supervision, Funding acquisition, Methodology, Writing—review and editing

### Author ORCIDs

Yuh Chwen G Lee, http://orcid.org/0000-0002-0081-7892
Gary H Karpen, http://orcid.org/0000-0003-1534-0385

## Additional files

### Major datasets

The following dataset was generated:

| Author(s) | Year | Dataset title | Dataset URL | Database, license, and accessibility information |
|---|---|---|---|---|
| Lee YCG, Karpen GH | 2017 | Pervasive epigenetic effects of Drosophila euchromatic transposable elements impact their evolution | https://www.ncbi.nlm.nih.gov/geo/query/acc.cgi?acc=GSE94742 | Publicly available at the NCBI Gene Expression Omnibus (accession no: GSE94742) |

The following previously published datasets were used:

| Author(s) | Year | Dataset title | Dataset URL | Database, license, and accessibility information |
|---|---|---|---|---|
| Langley C, Pool J | 2015 | 1000 Drosophila genomes | https://trace.ncbi.nlm.nih.gov/Traces/sra/?study=SRP006733 | Publicly available at the NCBI Sequence Read Archive (accession no: SRP006733) |
| Brennecke J, Hannon G | 2008 | Deep sequencing of Drosophila melanogaster small RNAs | https://www.ncbi.nlm.nih.gov/geo/query/acc.cgi?acc=GSE11086 | Publicly available at the NCBI Gene Expression Omnibus (accession no: |

GSE11086)

| Kofler R, Schlötterer C | 2015 | Massive bursts of TE activity in Drosophila | http://www.ebi.ac.uk/ena/data/view/PRJEB6673 | Publicly available at the EMBL-EBI European Nucleotide Archive (accession no: PRJEB6673) |

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
