## [Decision Letter]

Thank you for submitting your article "Pervasive epigenetic effects of *Drosophila* euchromatic transposable elements impact their evolution" for consideration by *eLife*. Your article has been favorably evaluated by Detlef Weigel (Senior Editor) and three reviewers, one of whom, Magnus Nordborg (Reviewer #1), is a member of our Board of Reviewing Editors. The following individual involved in review of your submission has agreed to reveal their identity: Brandon Gaut (Reviewer #2).

The reviewers have discussed the reviews with one another and the Reviewing Editor has drafted this decision to help you prepare a revised submission.

Summary:

This is a very thorough and important demonstration of the idea that epigenetic silencing of TEs in *Drosophila* may affect nearby genes, and lead to selection against TE insertions. This agrees with similar observations in plants, and suggests that this is a universal aspect of TE biology (at least in organisms with compact genomes).

Essential revisions:

All three reviewers are positive, but have produced lengthy, detailed, and non-overlapping lists of comments. Having gone through these comments, we think they are generally reasonable, self-explanatory and constructive, and we will therefore not follow standard *eLife* practice and try to extract the most important ones. You should obviously feel free to disagree with particular points, but in that case you will need to convince us that you are right!

*Reviewer #1:*

This is a really good paper that will likely prove to be major contribution to our understanding of TE dynamics. I have only a few concerns.

First, the paper is not up on the recent work in plants. The list quite long, but I would suggest starting with the following (and working backward from the refs):

Quadrana, Leandro, Amanda Bortolini Silveira, George F. Mayhew, Chantal LeBlanc, Robert A. Martienssen, Jeffrey A. Jeddeloh, and Vincent Colot. 2016. "The *Arabidopsis thaliana* Mobilome and Its Impact at the Species Level." *eLife* 5 (June). doi:10.7554/*eLife*.15716. [epigenetics silencing of TEs and selection on TEs in euchromatin]

Stuart, Tim, Steven Eichten, Jonathan Cahn, Yuliya Karpievitch, Justin Borevitz, and Ryan Lister. 2016. "Population Scale Mapping of Transposable Element Diversity Reveals Links to Gene Regulation and Epigenomic Variation." *eLife* 5 (December). doi:10.7554/*eLife*.20777. [epigentic silencing]

Kawakatsu, Taiji, Shao-Shan Carol Huang, Florian Jupe, Eriko Sasaki, Robert J. Schmitz, Mark A. Urich, Rosa Castanon, et al. 2016. "Epigenomic Diversity in a Global Collection of *Arabidopsis thaliana* Accessions." Cell 166 (2). Elsevier: 492-505. [genomewide epigenetics]

Note that the above look at DNA methylation, but this is essentially a proxy for H3K9.

Second, I'm somewhat concerned by using "present in Zambia" as a proxy for allele frequency. The logic is OK, but it is a bit crude, and there is a real chance that TE's outside Africa could have been affected by the environment. Also (subsection “TEs with epigenetic effects are more strongly selected against”, second paragraph), you do seem to have some real frequency data. Where is the figure?

Third, the last paragraph of the subsection “TEs with epigenetic effects are more strongly selected against”, about age and frequency is, frankly, very confused. A causal (statistical) relationship between age and frequency of a mutation is expected under any population genetics model except strong balancing selection. You observe an inverse correlation between silencing and frequency, and attribute this to purifying selection. I believe you are right, but it is formally possible that silencing simply decays with time. The right (and only) way to test this is to get an estimate of age that is independent of allele frequency. You could look for accumulation of mutations between copies of the same insertion, but you would never have enough power (I think – time is too short). Better is to look for the extent of haplotype sharing (the extent of linkage disequilibrium) surrounding insertion alleles. This decays with time, and alleles under selection should be surrounded by more extensive LD than neutral ones, conditional on allele frequency. This would be nice independent demonstration of selection on the more silenced ones.

The age of the family is not relevant to this (the time scales are completely different).

*Reviewer #2:*

In this remarkably thorough paper, Lee and Karpen examine the epigenetic effects of TE insertions on euchromatic regions of *Drosophila* genomes. The basic premise is that TEs elevate the incidence and spread of repressive epigenetic marks, causing dampening effects on the expression of nearby genes. Depending on the gene in question, these dampening effects could lead to deleterious selection on the TE-containing haplotypes. As such, epigenetic modifications of TEs may be a primary cause of deleterious selection against them. These ideas have been bandied about for a few years now, but there have been few thorough analyses of the idea.

To test these ideas, the authors start by comparing two inbred *Drosophila* strains for their TE content and the association of individual TE insertions (and euchromatic regions, for that matter) with enrichment of a repressive mark (H3K9me3). They then detail a number of analyses showing that TE insertions are associated with enrichment, that this enrichment often spreads to the TE's flanking regions, which this local enrichment can affect flanking genic regions, and that genic enrichment correlates negatively with gene expression. They also show that enrichment tends to be higher for TEs found at low frequency, which is a direct prediction of the 'deleterious selection by epigenetic modification' hypothesis. Finally, they extend comparisons to *D. simulans*. They present evidence that the epigenetic response is stronger in *D. simulans*, perhaps (?) due to modification of the titration of heterochromatic proteins.

This study is a welcome and important contribution to the field. It is novel in many aspects but particularly in its depth. The comparison between inbreds permit the address of existing ideas on a genome-wide scale but also permit comparisons among TE families, etc. In fact, I like this paper so much that I have no major comments about the content or any need for additional analyses. That said, it is a lengthy paper and, as might be expected of such a detailed work, there are many areas where clarity could be improved.

Specific comments:

Abstract: The abstract needs to be refined, as, in its current shape, it lacks in detail (e.g., H3K9me3).

Introduction: As a non-drosophilist I found myself puzzled by some of the statements about piRNAs and siRNAs that come later in the manuscript. My comprehension might have been helped by a paragraph or two that reviews known mechanisms of TE epigenetic modification in Drosophila. Would it be worth adding this information? [I was also puzzled about the mechanism of spread. If there is prior knowledge about this, it might be useful to add the information to a paragraph or two on mechanisms.]

Subsection “Euchromatic TEs exhibit extensive epigenetic effects on adjacent sequences”, fourth paragraph – because measures like the "extent of spread", "% increase in H3K9me2", "with/without epigenetic effect" etc. are used throughout, I think it is well worth giving their explicit definitions in the Results in addition to the Materials and methods, especially since the definitions are pretty short. This isn't necessary but may aid clarity for readers.

Subsection “TE families of LTR-type and targeted by piRNAs show stronger epigenetic effects”, third paragraph – I had difficulties synthesizing the statement that "there is no difference in the amount of piRNAs targeting different types of TE families", with the fact that piRNA amounts across families correlate with things like the "mean extent of spread" in Table 1. I guess this is saying that you can still rank families by piRNA amount and get some neat correlations, but it seems a bit strange.

Anyway, if nothing else, the description of Table 1 needs to be improved. From the text, I eventually realized these were Spearman correlations, but the table is not well described in its heading or in its first citation in the text. Finally, I believe this section (subsection “TE families of LTR-type and targeted by piRNAs show stronger epigenetic effects”) could benefit from some clarity about a couple of things: 1) how do piRNAs and siRNAs differ? 2) mapping piRNAs/siRNAs to TEs can be tricky due to multiple mapping phenomena and can totally mislead an analysis if not done carefully. Maybe I missed it in the Materials and methods, but it wasn't clear to me how the data in Table 1 were gathered (i.e., per family counts of piRNAs and siRNAs). Were pi/siRNAs counted once? Weighted by number of mapping positions? I assume they were mapped to the genome (as opposed to a reduced representation TE dataset). The variation in how these things are done is pretty huge; I just ask that it be clear. (Again, didn't see it in Materials and methods at all, but I think the Materials and methods implies that counts were gathered from another study?)

Subsection “TEs with epigenetic effects are more strongly selected against”, last paragraph – I found most of the discussion on age to be a bit muddled. If I'm not mistaken, frequency and age should covary, as argued, and (all else being equal) both should covary with epigenetic effect. Thus, the findings in the aforementioned paragraph seem to me to be unlikely, unless age is misestimated. My guess is that the sequence similarity measure to the reference is a misleading or noisy. (Depending on the history in a particular TE family, it's easy to envision cases where it might be positively misleading.) Not sure what to advise here, but I suspect it needs some more thought? Or could even be removed?

Subsection “Epigenetic effects of TEs result in differential epigenetic states of adjacent coding genes”, second paragraph – I found myself a bit confused here. Is the z-score compared between genes that differ in the presence/absence of flanking TEs in the two lines? If so, what window size defines "flanking"? Or is it really just the closest flanking TE without reference to distance? I'd just like a few more methodological details here in the flow of description of the results.

Table 2: Despite my best attempts, I could not understand it at all! Can you provide an example in the footnotes (for example, explicitly define the data in the first row) or in the text? Maybe I'm missing something obvious, but I just didn't get it!

Discussion, third paragraph: again, Table 2 is referred to almost obliquely. It'd be nice to be described in more detail or add a sentence that references the table explicitly – i.e., "For example, for TEs with higher enrichment in RAL315…". Great points about inbreds (Discussion, fourth paragraph) and synergistic epistasis (Discussion, fifth paragraph). Really enjoyed them!

Subsection “Chip-seq data analysis”, second paragraph and following. These are key definitions that (maybe) should be in the Results but definitely should be defined more clearly and precisely. I got the gist, but more precision and care are merited.

*Reviewer #3:*

In this paper, the authors analyze the effects that polymorphic TE insertions have on chromatin states in *Drosophila*. This paper is an extension of a previous study that was limited by the fact that the effect of polymorphic TE insertions could not be determined since only one reference was used (from ModEncode, for example). In this paper, the authors explicitly examine the effects of TE insertion polymorphism on variation in repressed chromatin signatures within the genome. This is an important contribution to the field and is very well written.

My major concern, which I will point out specifically in several places, is that the representation of the data limits the reader’s ability to understand the variation in the signal. In particular, many of the figures compress the data into box plots, where this compression is not appropriate. I will alert the authors to this paper:

"Beyond Bar and Line Graphs: Time for a New Data Presentation Paradigm". While the authors might think this concern is nitpicky, I don't think so. The authors are aiming for a high-profile paper and the readers should be able to see the structure of the underlying data and variation. Not all figures are at issue, but some can be revised.

The article is lacking reference to the work of the Kalmykova lab. In particular, the article must cite Shpiz et al. Euchromatic transposon insertions trigger […] This paper is an important contribution to the field.

Subsection “Euchromatic TEs exhibit extensive epigenetic effects on adjacent sequences”, second paragraph: It would be worth illuminating in the text the interesting observation in Figure 1 that embryos show these effects most strongly.

Subsection “Euchromatic TEs exhibit extensive epigenetic effects on adjacent sequences”, fourth paragraph: It is a little awkward to say "homologous sequences that lack nearby TEs." I think the authors mean "with corresponding alleles lacking the TE insertion."

Subsection “Euchromatic TEs exhibit extensive epigenetic effects on adjacent sequences”, fourth paragraph: Since the metrics "extent of spread" and "% increase in H3K9me" are used throughout, please exactly define these metrics here, not just in the Methods.

Subsection “Euchromatic TEs exhibit extensive epigenetic effects on adjacent sequences”, fourth paragraph: Please exactly define the characteristic: "associated with enrichment for H3K9me2 in at least 1 kb". Likewise, please define how the extent of spreading is defined (what is the cutoff for spread).

Figure 2 legend. Please clarify in the legend that these are TE insertion alleles that are absent in the other strain.

Subsection “TE families of LTR-type and targeted by piRNAs show stronger epigenetic effects”: There is a suggestion that LTRs have stronger effects. This is demonstrated in several ways, but I don't follow the statistics:

1) Subsection “TE families of LTR-type and targeted by piRNAs show stronger epigenetic effects”, second paragraph: There is suggestion of enrichment with several statements such as "eight of 11 TE families with over half of insertions […] are LTR-type" Is this enrichment significant, given there are more LTR families? A Fisher's exact would be suitable here. A similar point can be made about the next sentence.

2)They perform a statistical test of this phenomenon (subsection “TE families of LTR-type and targeted by piRNAs show stronger epigenetic effects”, second paragraph), but it is not clear the test is appropriate. Here, they seem to be binning all LTRs (I could be wrong) but this Mann-Whitney U result may be solely driven by one abundant LTR family, such as *copia* or *roo*. The authors should provide a more convincing case that LTRs really exert a stronger effect, in general, instead of this being driven by some families. This could be done in an ANOVA framework, where the effect of TE class is explicitly tested, in contrast to family level effects.

Subsection “TE families of LTR-type and targeted by piRNAs show stronger epigenetic effects”, third paragraph: "It is worth noting […]" I don't understand this sentence. It appears that they are saying that there are no differences, between strains, in piRNA abundance for each given family. Is that correct? Then why are P-values indicated for two genotypes, sense such a test between two strains would have one P value?

Figure 5 is an example of data compression with bar charts that is not appropriate. First, frequency should not be binned in this manner. Such binning is arbitrary and masks the underlying data. Please plot these values continuously. In addition, for 5B and 5C, plot the Y axis continuously for each TE insertion.

Also, in regard to the analysis of population frequencies, this analysis seems completely confounded by the previously reported family level effects and *copia* number effects. For example, *copia* elements (LTRs, with large *copia* number) may also have low frequencies for other reasons that have nothing to do with the effect they exert on local chromatin. I imagine there are two ways to deal with this. 1) All contrasts must control for within family effects. For example, the test should be: Do *copia* elements with a stronger chromatin effect have lower frequencies than other *copia* elements? Do *roo* elements with a stronger chromatin effect have lower frequencies than other *roo* elements? The family level effects must be accounted for. 2) A formal way to do this would be to directly model population frequency and perform a multiple logistic regression where family effects and chromatin effects are evaluated for their capacity to predict frequencies in the population. This way, family effects and chromatin effects can be explicitly measured and evaluated.

Right now, this binning approach is not appropriate.

Subsection “Epigenetic effects of TEs result in differential epigenetic states of adjacent coding genes”. Similar data compression by binning for the expression analysis is not appropriate. The Fisher's exact test approach in Table 2 doesn't seem appropriate. There is a very nice paired structure to these data that can simply be measured as the difference in RPKM between alleles with a TE and alleles without a TE. Then, a scatter plot can be shown to demonstrate the degree to which chromatin effects are predictive of gene expression differences. Here, the X-axis would be some continuous measure of chromatin effect and the Y-axis would be the measured effect on gene expression. Such a scatter plot is essential for the reader to evaluate the nature of these data. These bins do not allow the reader to understand what is going on.

Figure 7. It seems a binomial test would show 7B (proportion of TE spread) significant, but not for the rest of 7B. Please provide P values for a test of more dots above the diagonal than below the diagonal in 7B.

[Editors' note: further revisions were requested prior to acceptance, as described below.]

Thank you for resubmitting your work entitled "Pervasive epigenetic effects of *Drosophila* euchromatic transposable elements impact their evolution" for further consideration at *eLife*. Your article has been favorably evaluated by Detlef Weigel (Senior Editor) and three reviewers, one of whom is a member of our Board of Reviewing Editors.

The manuscript has been improved but there are some remaining issues that need to be addressed before acceptance, as outlined below:

As requested by Reviewer #1, please integrate other previous findings better.

As requested by Reviewer #3, please address points 1, 3, and 4 (or explain why this is not a good idea).

The paper is excellent: we just think it would be further improved with relatively little effort.

*Reviewer #1:*

I'm happy with your response to my comments with one exception: just adding a bunch of plant references at one point in the Introduction is not enough. Sure, the authors get the citation count credit they deserve, but in the name of scholarship you should do better.

What struck me when I read your paper is how closely it parallels findings in plants, and I think your paper would improve if you made this clear.

For example, the spreading of silencing has been shown several times in plants, and there is also strong evidence that selection depends on this. Population frequencies appear to be low, and the recent *Arabidopsis* paper by Colot's group clearly showed that insertions are uniform, but that selection removes them so that older (more frequent) TEs cluster around the centromeres.

I know it is a pain to do this, but there are several places in both Introduction and Discussion where you could note these similarities. It would really improve the paper, and would be a service to a field that is too often is organism-limited.

Someone needs to spend a day in a coffee house with a (virtual) stack of papers and a laptop. This is not too much to ask.

With respect to estimating the age of TEs, I understand that you cannot sequence the interior of the TE, but did you try the haplotype sharing? I think it could be more powerful than you suggest. This is real work, however, and I'm fine with your dropping the section instead.

*Reviewer #2:*

The authors have done a thorough job responding to the reviewers' comments. I heartily endorse publication in *eLife*.

*Reviewer #3:*

This is the second version of this article that I have read. The results are very interesting and important, but I feel the argument simplifies some complexity that should be more acknowledged. The authors have addressed many of my concerns, but, as in my first review, I will ask that the authors improve the presentation to allow readers to have a more clear understanding of the underlying complexities.

1) With regard to the definition of extent of spread, I am still slightly confused. The authors do a good job and provide a graphical representation of these definitions, but by "farthest window", how do they deal with values on one side being larger than values on the other side. Suppose the extent of spread to the left is 15 kb and the extent of spread to the right is 6 kb. Is a max function used such that 15 kb is the defined extent? Please be exact and precise in this definition.

2) Introduction, last paragraph, and elsewhere, I think there should be a bit more of a hedge on the conclusion that selection is determining the patterns. Therefore, a sentence like "we observed stronger selection against TE insertions" would preferably be "we find evidence that stronger selection against TE insertions"

3) This is my most significant comment. I am still confused about the expression analysis.

First, from Table 3, it appears this effect is only observed in one strain. In fact, for TEs in RAL315 (top set of genes in Table 2), the odds ratio is 0.95, the opposite expected. Therefore, it is not clear to me that one can argue that this result is straightforward. The authors should be upfront about this strain effect. It's interesting! But it certainly makes the story more complex and reduces the strength of their broader conclusion.

Second, Table 3 can still be improved. For one, the Fisher's Exact test is being performed against the background genes (with respect to sign of effect of enrichment and expression between the two strains), but the way it is presented, the nature of values in the 2X2 table is cryptic. From first look, one might think that the four values in each row are members of the 2X2 table. Now, this is not the correct interpretation, but I encourage the authors to make this clearer. In particular, why not just provide the four different 2X2 tables? This is also made confusing from the fact that, it appears, the numbers are wrong in the example below the table. By my count, for the RAL315 TEs, 56 is the value in the focal cell (H3K9me2 higher in 315, Higher expression in 360), but the values of the other cells (based on the text below) should sum to 121, correct? But, by my count, the rest of the genes with the other states add to 177 (75+56+46). I am sorry if I am wrong, but I guess I am still confused. I do think that presentation of the actual 2X2 tables that go into the FET would be best.

In light of these issues, I am still not convinced that Fisher's exacts tests of this kind are the best way to present the data. In my last review, I had requested a figure more like Figure 7 appreciate the authors providing it. However, it is clear from this new figure that the effect that these TEs have on expression, via H3K9me2, is complicated. For example, from the scatter plots for RAL315 TE insertions, there are 131 genes with TE inserts that have higher H3K9me2 in RAL315. However, among these genes, most (73) show a higher level of expression in 315 – the opposite pattern expected. It is also appears from these scatterplots that there is perhaps no correlation between fold H3K9me2 enrichment differences and differences in expression. This is a very interesting and extremely important complexity that must be made evident to the reader. Therefore, I request that these scatter plots be provided in the main figures with a revised version of Table 3 correlation coefficient be provided for these scatter plots.

4) The authors have done a good job controlling for the non-independent effects of family and spread of H3K9me in predicting TE insertion frequency. This is evident in their Methods (subsection “TE population frequency analysis”). However, Table 2 doesn't provide the jointly estimated regression coefficient for family. It just provides regression coefficients and p-values for extent and magnitude of spread. Please provide results from the full models, where regression coefficients were jointly estimated for epigenetic and family effects.

---

## [Author Response]

*Reviewer #1:*

*This is a really good paper that is likely to prove to be major contribution to our understanding of TE dynamics. I have only a few concerns.*

*First, the paper is not up on the recent work in plants. The list quite long, but I would suggest starting with the following (and working backward from the refs):*

*Quadrana, Leandro, Amanda Bortolini Silveira, George F. Mayhew, Chantal LeBlanc, Robert A. Martienssen, Jeffrey A. Jeddeloh, and Vincent Colot. 2016. "The Arabidopsis thaliana Mobilome and Its Impact at the Species Level." eLife 5 (June). doi:10.7554/eLife.15716. [epigenetics silencing of TEs and selection on TEs in euchromatin]*

*Stuart, Tim, Steven Eichten, Jonathan Cahn, Yuliya Karpievitch, Justin Borevitz, and Ryan Lister. 2016. "Population Scale Mapping of Transposable Element Diversity Reveals Links to Gene Regulation and Epigenomic Variation." eLife 5 (December). doi:10.7554/eLife.20777. [epigentic silencing]*

*Kawakatsu, Taiji, Shao-Shan Carol Huang, Florian Jupe, Eriko Sasaki, Robert J. Schmitz, Mark A. Urich, Rosa Castanon, et al. 2016. "Epigenomic Diversity in a Global Collection of Arabidopsis thaliana Accessions." Cell 166 (2). Elsevier: 492-505. [genomewide epigenetics]*

*Note that the above look at DNA methylation, but this is essentially a proxy for H3K9.*

We added references suggested by the reviewer along with other plant references. These changes can be in the third paragraph of the Introduction.

*Second, I'm somewhat concerned by using "present in Zambia" as a proxy for allele frequency. The logic is OK, but it is a bit crude, and there is a real chance that TE's outside Africa could have been affected by the environment. Also (subsection “TEs with epigenetic effects are more strongly selected against”, second paragraph), you do seem to have some real frequency data. Where is the figure?*

We did estimate the frequency of individual TEs in the Zambian population. Consistent with previous findings that most of the TEs are rare in *Drosophila,* the majority (68.5%) of the TEs were not detected in the Zambian population. We thus used either (1) TE’s population frequencies (Spearman rank correlation tests, pointed out by reviewer) or (2) TE insertion present in Zambian population or not (Mann-Whitney U test), to investigate the associations between TE’s epigenetic effects and population frequencies. We further revised our text to clarify this point (subsection “TEs with epigenetic effects are more strongly selected against”, second paragraph). However, we agree with reviewers 1 and 3 that it is necessary to show the full-range of variation in TE’s population frequencies, so we added additional supplementary figures (Figure 5—figure supplement 1 and Figure 5—figure supplement 3).

The two strains used in the study are from DGRP, a collection of *D. melanogaster* strains from North America. We chose the Zambian population to estimate population frequencies of TEs, instead of the North American population, for the following reasons. One is regarding the demographic history of this population, which we discuss in the main text (subsection “TEs with epigenetic effects are more strongly selected against”, second paragraph). Others are technical reasons. The sequencing of DGRP genomes used a mixture of 454 and pair-end Illumina of several read lengths (36bp-125bp). The sequencing coverage is also highly variable between genomes. The power to detect TE insertions is thus highly variable between strains in the DGRP panel. In addition, there are extensive identity-by-descent (IBD) regions among these genomes (Cridland et al. 2013 MBE), which would further complicate the analysis. In contrast, the Zambian population was sequenced with the same sequencing technology, similar read-length and read depth, while having limited IBD regions (Lack et al. 2015 Genetics). It is worth noting that the two DGRP strains used in the study were sequenced with pair-end Illumina long read lengths (125bp) and with high sequencing coverage. There is also limited IBD identified between these two strains.

*Third, the last paragraph of the subsection “TEs with epigenetic effects are more strongly selected against”, about age and frequency is, frankly, very confused. A causal (statistical) relationship between age and frequency of a mutation is expected under any population genetics model except strong balancing selection. You observe an inverse correlation between silencing and frequency, and attribute this to purifying selection. I believe you are right, but it is formally possible that silencing simply decays with time. The right (and only) way to test this is to get an estimate of age that is independent of allele frequency. You could look for accumulation of mutations between copies of the same insertion, but you would never have enough power (I think – time is too short). Better is to look for the extent of haplotype sharing (the extent of linkage disequilibrium) surrounding insertion alleles. This decays with time, and alleles under selection should be surrounded by more extensive LD than neutral ones, conditional on allele frequency. This would be nice independent demonstration of selection on the more silenced ones.*

*The age of the family is not relevant to this (the time scales are completely different).*

We agreed with reviewers 1 and 2 that our original attempt to exclude age of TE insertions as a contributor to variation in TE’s population frequencies is not well discussed and is confusing.

We really like the analysis the reviewer suggested. However, because these genomes were sequenced with Illumina short reads, we are unable to gather the internal sequences of individual TE insertions and would detect very few mutations, if any, for inference (on top of the issue of short time scale). Given that most TEs have very low population frequencies or appear as singletons (which are of particular interests) in our analysis, using haplotype sharing to infer age of TE insertions is expected to have limited statistical power. We anticipate future long-read sequencing will enable us to assemble the internal sequence of individual TE insertions, which will allow using sequence comparisons (e.g. divergence between two LTRs of a LTR insertion, or more internal sequences) to infer TE age. Because we are unable to provide unequivocal data with respect to this issue, we removed this section of the Discussion.

*Reviewer #3: […] My major concern, which I will point out specifically in several places, is that the representation of the data limits the readers ability to understand the variation in the signal. In particular, many of the figures compress the data into box plots, where this compression is not appropriate. I will alert the authors to this paper:*

*"Beyond Bar and Line Graphs: Time for a New Data Presentation Paradigm". While the authors might think this concern is nitpicky, I don't think so. The authors are aiming for a high-profile paper and the readers should be able to see the structure of the underlying data and variation. Not all figures are at issue, but some can be revised.*

We thank the reviewer for pointing out this thoughtful paper. To ensure concise presentation of the data, we retained several boxplots and a table in the main text, while added additional supplementary figures that allow readers to evaluate the full variation of the data. Please also see our response to comments on specific figures below.

*The article is lacking reference to the work of the Kalmykova lab. In particular, the article must cite Shpiz et al. Euchromatic transposon insertions trigger […]. This paper is an important contribution to the field.*

We agree with the reviewer and added discussion of this reference to the Discussion section (third paragraph).

*Subsection “Euchromatic TEs exhibit extensive epigenetic effects on adjacent sequences”, second paragraph: It would be worth illuminating in the text the interesting observation in Figure 1 that embryos show these effects most strongly.*

We added discussions about this observation as suggested (subsection “Euchromatic TEs exhibit extensive epigenetic effects on adjacent sequences”, first paragraph).

*Subsection “Euchromatic TEs exhibit extensive epigenetic effects on adjacent sequences”, fourth paragraph: It is a little awkward to say "homologous sequences that lack nearby TEs." I think the authors mean "with corresponding alleles lacking the TE insertion."*

We revised our text to improve clarity (subsection “Euchromatic TEs exhibit extensive epigenetic effects on adjacent sequences”, third paragraph).

*Subsection “Euchromatic TEs exhibit extensive epigenetic effects on adjacent sequences”, fourth paragraph: Since the metrics "extent of spread" and "% increase in H3K9me" are used throughout, please exactly define these metrics here, not just in the Methods.*

*Subsection “Euchromatic TEs exhibit extensive epigenetic effects on adjacent sequences”, fourth paragraph: Please exactly define the characteristic: "associated with enrichment for H3K9me2 in at least 1 kb". Likewise, please define how the extent of spreading is defined (what is the cutoff for spread).*

We agree with reviewer 3, and now include a figure to more clearly define all three metrics (“associated with enrichment for H3K9me2 in at least 1 kb”, "extent of spread" and "% increase in H3K9me") (subsection “Euchromatic TEs exhibit extensive epigenetic effects on adjacent sequences”, third paragraph), in addition to added text in the Results and revised Materials and methods (subsection “ChIP-Seq data analysis”, second paragraph).

*Figure 2 legend. Please clarify in the legend that these are TE insertion alleles that are absent in the other strain.*

We revised the figure legend accordingly.

*Subsection “TE families of LTR-type and targeted by piRNAs show stronger epigenetic effects”: There is a suggestion that LTRs have stronger effects. This is demonstrated in several ways, but I don't follow the statistics:*

*1) Subsection “TE families of LTR-type and targeted by piRNAs show stronger epigenetic effects”, second paragraph: There is suggestion of enrichment with several statements such as "eight of 11 TE families with over half of insertions…are LTR-type" Is this enrichment significant, given there are more LTR families? A Fisher's exact would be suitable here. A similar point can be made about the next sentence.*

These descriptions were intended as a general summary of the TE family data, and not as direct, quantitative support for the conclusion that LTR-type TE families have stronger epigenetic effects. We refrain from doing the Fisher’s Exact test because the categorization is fairly arbitrary (e.g. TE families with >5kb average spread of H3K9me2). Accordingly, we revised the text to avoid misleading readers at this stage of the analysis (subsection “TE families of LTR-type and targeted by piRNAs show stronger epigenetic effects”, second paragraph), and only make a more definitive conclusion after quantitative analysis using Mann-Whitney U tests(see response to next comment).

*2) They perform a statistical test of this phenomenon (subsection “TE families of LTR-type and targeted by piRNAs show stronger epigenetic effects”, second paragraph), but it is not clear the test is appropriate. Here, they seem to be binning all LTRs (I could be wrong) but this Mann-Whitney U result may be solely driven by one abundant LTR family, such as copia or roo. The authors should provide a more convincing case that LTRs really exert a stronger effect, in general, instead of this being driven by some families. This could be done in an ANOVA framework, where the effect of TE class is explicitly tested, in contrast to family level effects.*

The Mann-Whitney U tests were comparing “the proportion of TEs with epigenetic effects”, “the average extent of H3K9me2 spread”, and “average% increase in H3K9me2 enrichment” between LTR-type TE families and other types of TE families. In other words, one TE family is one data point. So, *roo and copia,* despite being abundant, are two data points in these analyses and are unlikely to be the only TE families driving the significant patterns. We agree that the “units” used for these comparisons was unclear and thus revised the text (subsection “TE families of LTR-type and targeted by piRNAs show stronger epigenetic effects”, second paragraph).

*Subsection “TE families of LTR-type and targeted by piRNAs show stronger epigenetic effects”, third paragraph: "It is worth noting […]" I don't understand this sentence. It appears that they are saying that there are no differences, between strains, in piRNA abundance for each given family. Is that correct? Then why are P-values indicated for two genotypes, sense such a test between two strains would have one P value?*

What we meant is “irrespective of the genotype (*wK* or *w1118*), piRNA abundance is not significantly different between LTR-type and other type of TE families”. Instead of abbreviating the two p-values as “*p* < 0.19”, we now included p-values for each genotype to avoid this confusion. We also revised the text accordingly (subsection “TE families of LTR-type and targeted by piRNAs show stronger epigenetic effects”, third paragraph).

*Figure 5 is an example of data compression with bar charts that is not appropriate. First, frequency should not be binned in this manner. Such binning is arbitrary and masks the underlying data. Please plot these values continuously. In addition, for 5B and 5C, plot the Y axis continuously for each TE insertion.*

We agree with reviewers 1 and 3 that the binning may not seem appropriate without knowledge of the variation of the data. Consistent with previous findings that most of the TEs are rare in *Drosophila,* the majority of the TEs (68.5%) were not detected in the Zambian population, and most of other TE insertions have low population frequencies. Accordingly, we performed two types of analyses – one by categorizing TEs into ‘observed and not observed’ in the Zambian population, and the other using the continuous distribution of TE frequencies. We have revised the text to clarify this point (subsection “TEs with epigenetic effects are more strongly selected against”, second paragraph). As recommended by the reviewer, we added new supplementary figures (Figure 5—figure supplement 1, Figure 5—figure supplement 2, Figure 5—figure supplement 3) that show the full distribution of TE frequencies. Please also see our response to the second comment of reviewer 1.

*Also, in regard to the analysis of population frequencies, this analysis seems completely confounded by the previously reported family level effects and copia number effects. For example, copia elements (LTRs, with large copia number) may also have low frequencies for other reasons that have nothing to do with the effect they exert on local chromatin. I imagine there are two ways to deal with this. 1) All contrasts must control for within family effects. For example, the test should be: Do copia elements with a stronger chromatin effect have lower frequencies than other copia elements? Do roo elements with a stronger chromatin effect have lower frequencies than other roo elements? The family level effects must be accounted for. 2) A formal way to do this would be to directly model population frequency and perform a multiple logistic regression where family effects and chromatin effects are evaluated for their capacity to predict frequencies in the population. This way, family effects and chromatin effects can be explicitly measured and evaluated.*

*Right now, this binning approach is not appropriate.*

We performed regression analysis to jointly consider the effects of TE’s epigenetic effects and family identity on their population frequencies. To ensure that our conclusion is robust with respect to the assumed regression models, we modeled the response variable (TE’s population frequencies) with several types of regression analyses.

1) Because a large proportion of TEs are not observed in the Zambian population (68.5%), we modeled TE’s population frequencies as a dichotomous variable and performed logistic regression.

2) Because the population frequencies of TEs were first observed as “counts” (the number of individuals with a particular TE insertion in the population), and the distribution of TE frequencies is skewed towards small numbers (L-shape), TE’s population frequencies can be treated as Poisson distributed. However, the variance of TE’s population frequencies is larger than the mean (overdispersion). Accordingly, we modeled TE frequencies as either “quasi-poisson” or “negative binomial”, since both are usually used for modeling overdispersed Poisson-like count variables, and performed generalized linear regression.

Because we do not have *a priori* knowledge about whether the variation in population frequencies between TE families is due to a fixed effect (i.e. each TE family has biological properties that make its insertions always have lower/higher population frequencies) or a random effect (variation between families), we treated the influence of TE family identity on TE’s population frequencies either as a fixed effect (generalized linear model) or a random effect (generalized mixed linear model).

For all regression models, we observed negative associations between TE’s epigenetic effects and population frequencies, after accounting for the effect of TE family identity. The regression coefficients are statistically significant for majority of the models considered, further corroborating the finding that TEs with stronger epigenetic effects are more likely to have low population frequencies. These new results are now in Table 2 (subsection “TEs with epigenetic effects are more strongly selected against”, third paragraph).

*Subsection “Epigenetic effects of TEs result in differential epigenetic states of adjacent coding genes”. Similar data compression by binning for the expression analysis is not appropriate. The Fisher's exact test approach in Table 2 doesn't seem appropriate. There is a very nice paired structure to these data that can simply be measured as the difference in RPKM between alleles with a TE and alleles without a TE. Then, a scatter plot can be shown to demonstrate the degree to which chromatin effects are predictive of gene expression differences. Here, the X-axis would be some continuous measure of chromatin effect and the Y-axis would be the measured effect on gene expression. Such a scatter plot is essential for the reader to evaluate the nature of these data. These bins do not allow the reader to understand what is going on.*

We now included a supplementary figure describing the full variation of the data (Figure 6—figure supplement 2). As mentioned in the Discussion, for various reasons, we observe genes that are “not following” the prediction (e.g. the allele with a TE has high H3K9me2 enrichment but also high RNA transcript level). We thus performed Fisher’s Exact test, and the proportion of genes following the expected influence of TE insertion (i.e. alleles adjacent to TE insertions have higher H3K9me2 enrichment and lower transcript levels) is significantly more than expectation calculated from genes without TEs in 10kb.

*Figure 7. It seems a binomial test would show 7B (proportion of TE spread) significant, but not for the rest of 7B. Please provide P values for a test of more dots above the diagonal than below the diagonal in 7B.*

To test for differences in epigenetic effects of TEs between *D. melanogaster* and *D. simulans,* we used *paired* Wilcoxon tests (Wilcoxon signed-rank test) to assess significance in Figure 7. This is a nonparametric test that takes into account both the sign (i.e. the number of dots above/below the diagonal in Figure 7) and the magnitude of differences. In contrast, a binomial test would miss the important information in the “magnitude of differences”. Accordingly, we still presented our original results using Wilcoxon signed-rank test.

[Editors' note: further revisions were requested prior to acceptance, as described below.]

*Reviewer #1:*

*I'm happy with your response to my comments with one exception: just adding a bunch of plant references at one point in the Introduction is not enough. Sure, the authors get the citation count credit they deserve, but in the name of scholarship you should do better.*

*What struck me when I read your paper is how closely it parallels findings in plants, and I think your paper would improve if you made this clear.*

*For example, the spreading of silencing has been shown several times in plants, and there is also strong evidence that selection depends on this. Population frequencies appear to be low, and the recent Arabidopsis paper by Colot's group clearly showed that insertions are uniform, but that selection removes them so that older (more frequent) TEs cluster around the centromeres.*

We agree that clearer acknowledgement of the relevant plant literature is appropriate, and have revised the text accordingly. However, we are confused about the Quadrana et al. 2016 (study from Colot’s group); we were unable to find data that demonstrated selection against spreading of methylation from TEs. They found that 10% of TE insertion sites have spreading of methylation in most accessions, but this includes accessions without TE insertions. These tend to be in pericentromeric regions, thus the enrichment of DNA methylation likely results from processes other than TE-induced spreading of this epigenetic mark. If this is not what the reviewer was referring to, we would appreciate the reviewer pointing out the relevant data presented in Quadrana et al. 2016. In addition, the accumulation of TEs in pericentromeric regions could also result from lower probability of ectopic exchange and/or reduced efficacy of selection (Hill-Robertson effects). Nevertheless, in this revised manuscript we still incorporate other important observations in this study (see below).

It is worth noting that, to avoid potential confounding influence of pericentromeric and peritelomeric regions already enriched for repressive epigenetic marks (likely the situation for *Arabidopsis* studies), we used highly conservative euchromatin-heterochromatin boundaries and only analyzed TEs and sequences that are in euchromatic regions.

Interestingly, similar to our observations, the two recent studies in *Arabidopsis* that jointly analyzed mobilomes and transcriptomes (Quadrana et al. 2016 and Stuart et al. 2016) also found complex influences of TEs on differential gene expression (also see below). However, neither study further associated these data with epigenomic data. Accordingly, we were unable to further contrast our observations with the *Arabidopsis* literature.

*I know it is a pain to do this, but there are several places in both Introduction and Discussion where you could note these similarities. It would really improve the paper, and would be a service to a field that is too often is organism-limited.*

*Someone needs to spend a day in a coffee house with a (virtual) stack of papers and a laptop. This is not too much to ask.*

We agree that more detailed discussion of the rich plant literature would provide broader context for our study, and we revised several places in Introduction and Discussion accordingly (see list below for details). We wholeheartedly agree that the field is often too organism-centric, and hope this revised manuscript can contribute to breaking that barrier.

1) In Introduction (third paragraph) and Discussion (second paragraph), we discussed that, in *Arabidopsis*, TEs have been identified as a major cause for differential epigenetic states between genomes, and some of the recent studies further demonstrated that is due to spreading of repressive epigenetic marks from silenced TEs. 4

2) In Discussion (second paragraph), we compared the extent of spreading of repressive epigenetic marks from TEs between *Drosophila* and *Arabidopsis*, which is on average more extensive in the former than the latter.

3) In Discussion (fourth paragraph), we pointed that our observed complexity of TEs’ influence on the differential gene expression echoes those documented in *Arabidopsis*.

4) In Discussion (seventh paragraph, already discussed in earlier version of the manuscript), we discussed the surprising parallel of associations between strength of epigenetic silencing of TEs and genomic TE content between closely related species in *Drosophila* and *Arabidopsis*.

It’s too bad that we don’t have coffee houses as lovely as those in Vienna, but hope we still did a proper job on this issue!

*With respect to estimating the age of TEs, I understand that you cannot sequence the interior of the TE, but did you try the haplotype sharing? I think it could be more powerful than you suggest. This is real work, however, and I'm fine with your dropping the section instead.*

We agree that haplotype sharing could be a really powerful approach to estimate the age of an allele. However, most of the TEs appear as singletons in *Drosophila* populations and there would be limited information for inference. An alternative approach may be to use haplotype sharing between alleles with and without TEs, which some groups are current developing. This is something of continuing interest, and we intend to formally address this in the near future (maybe with advice from the reviewer!). For this study, we would like to leave it out for the moment.

*Reviewer #3:*

*This is the second version of this article that I have read. The results are very interesting and important, but I feel the argument simplifies some complexity that should be more acknowledged. The authors have addressed many of my concerns, but, as in my first review, I will ask that the authors improve the presentation to allow readers to have a more clear understanding of the underlying complexities.*

*1) With regard to the definition of extent of spread, I am still slightly confused. The authors do a good job and provide a graphical representation of these definitions, but by "farthest window", how do they deal with values on one side being larger than values on the other side. Suppose the extent of spread to the left is 15 kb and the extent of spread to the right is 6 kb. Is a max function used such that 15 kb is the defined extent? Please be exact and precise in this definition.*

When the farthest windows are different between the left and right sides of the TE insertion, we used the smaller value in order to be conservative. In this example, the TE would be estimated to have 6kb spread. We clarified our definition in Materials and methods (subsection “ChIP-Seq data analysis”, second paragraph).

*2) Introduction, last paragraph, and elsewhere, I think there should be a bit more of a hedge on the conclusion that selection is determining the patterns. Therefore, a sentence like "we observed stronger selection against TE insertions" would preferably be "we find evidence that stronger selection against TE insertions"*

We revised the text in the last paragraph of the Introduction according to reviewer’s suggestion. We also double-checked that, at other places where selection was discussed, we did state that it is our evidence suggesting stronger selection, instead of directly observed stronger selection against TE insertions.

*3) This is my most significant comment. I am still confused about the expression analysis.*

*First, from Table 3, it appears this effect is only observed in one strain. In fact, for TEs in RAL315 (top set of genes in Table 2), the odds ratio is 0.95, the opposite expected. Therefore, it is not clear to me that one can argue that this result is straightforward. The authors should be upfront about this strain effect. It's interesting! But it certainly makes the story more complex and reduces the strength of their broader conclusion.*

*Second, Table 3 can still be improved. For one, the Fisher's Exact test is being performed against the background genes (with respect to sign of effect of enrichment and expression between the two strains), but the way it is presented, the nature of values in the 2X2 table is cryptic. From first look, one might think that the four values in each row are members of the 2X2 table. Now, this is not the correct interpretation, but I encourage the authors to make this clearer. In particular, why not just provide the four different 2X2 tables? This is also made confusing from the fact that, it appears, the numbers are wrong in the example below the table. By my count, for the RAL315 TEs, 56 is the value in the focal cell (H3K9me2 higher in 315, Higher expression in 360), but the values of the other cells (based on the text below) should sum to 121, correct? But, by my count, the rest of the genes with the other states add to 177 (75+56+46). I am sorry if I am wrong, but I guess I am still confused. I do think that presentation of the actual 2X2 tables that go into the FET would be best.*

This is an error at our end and we have corrected it (also see below).

*In light of these issues, I am still not convinced that Fisher's exacts tests of this kind are the best way to present the data. In my last review, I had requested a figure more like Figure 7 appreciate the authors providing it. However, it is clear from this new figure that the effect that these TEs have on expression, via H3K9me2, is complicated. For example, from the scatter plots for RAL315 TE insertions, there are 131 genes with TE inserts that have higher H3K9me2 in RAL315. However, among these genes, most (73) show a higher level of expression in 315 – the opposite pattern expected. It is also appears from these scatterplots that there is perhaps no correlation between fold H3K9me2 enrichment differences and differences in expression. This is a very interesting and extremely important complexity that must be made evident to the reader. Therefore, I request that these scatter plots be provided in the main figures with a revised version of Table 3 correlation coefficient be provided for these scatter plots.*

We agree with the reviewer that the complexity between the associations of H3K9me2 enrichment and transcript levels is of biological significance and should be more clearly pointed out. To address reviewer’s comments, we did the following in this revision.

1) For differential expression analysis, we excluded genes that are deemed to have no or extremely low expression by the modEncode developmental time course data. These genes indeed have no or extremely few mapped reads in our dataset, and the estimated expression fold changes are expected to be more prone to errors than other genes. Details are included in Materials and methods (subsection “RNA-seq analysis”, first paragraph).

2) Figure 3—figure supplement is now Figure 7. We replaced the confusing Table 3 with Figure 7, which includes 2x2 tables for testing whether there is an enrichment of genes supporting the epigenetic effects of TEs on gene expression. We also pointed out genes that are not following the expected trend (Figure 7, arrows; subsection “Epigenetic effects of TEs result in differential epigenetic states of adjacent coding genes”, last paragraph; Discussion, fourth paragraph) and the differences between two strains (subsection “Epigenetic effects of TEs result in differential epigenetic states of adjacent coding genes”, last paragraph), as pointed out by the reviewer.

3) As requested by the reviewer, we provided results for the correlation tests and (if the test is significant) the correlation coefficient between z score for differential epigenetic states and fold change for expression difference (subsection “Epigenetic effects of TEs result in differential epigenetic states of adjacent coding genes”, last paragraph). We found a negative, though weak, correlation between z score and log2 fold change for genes without TEs, but no significant correlations for genes with 10kb to TEs.

4) We provided potential explanations for the complex relationship between H3K9me2 enrichment status and differential gene expression.

A) Incidences of TE insertions leading to both down- and up-regulation of gene expression have been observed. Consistently, recent genome-wide studies in *Arabidopsis* found that TEs result in equal frequency of increased and decreased expression of neighboring genes (Discussion, fourth paragraph).

B) Structural variants may also contribute to differential gene expression (discussed in previous version of the manuscript).

C) This study used inbred strains, and TEs inducing strong reduction in transcript levels are expected to have been removed during the inbreeding process. The small sample size in our current study (two strains) likely precluded detecting subtle functional consequences of TE’s epigenetic effects that are expected prevalent in inbred strains (Discussion, fifth paragraph).

D) There are known exceptions to the negative influence of enrichment of heterochromatic marks on gene expression. In *Drosophila*, multiple essential genes are located in pericentromeric heterochromatin, and their expression depends on a heterochromatin environment (i.e. enrichment of H3K9me2/3 marks and heterochromatin proteins). Furthermore, several homologs of *D. melanogaster* heterochromatic genes are located in euchromatic regions in other species and, even in euchromatin, these homologs are also enriched with heterochromatin proteins/marks, suggesting conservation of epigenetic environment for gene function (Discussion, fourth paragraph).

*4) The authors have done a good job controlling for the non-independent effects of family and spread of H3K9me in predicting TE insertion frequency. This is evident in their Methods (subsection “TE population frequency analysis”). However, Table 2 doesn't provide the jointly estimated regression coefficient for family. It just provides regression coefficients and p-values for extent and magnitude of spread. Please provide results from the full models, where regression coefficients were jointly estimated for epigenetic and family effects.*

These data are now provided as [Supplementary-material SD2-data]. Because of the large number of TE families (which generates an extensive table), we did not provide this in the main text Table 2.